# FTP: Efficient Prefilling for Long-Context LLM Inference via FFN Token Pruning

## Abstract

Large Language Models (LLMs) have demonstrated remarkable performance across various NLP tasks, and have extended their capability to long-context scenarios. However, the increasing context length leads to longer inference time in both the prefilling and decoding stages. Existing token pruning methods primarily evict tokens to compress the KV cache, and only accelerate the decoding stage. Recent studies have extended token pruning to both stages, but they either yield subtle speedup during the prefilling stage or defer a portion of computations to the decoding phase. Critically, these approaches prioritize the attention module, overlooking the significant computations in the Feed-Forward Network (FFN) module.

In this work, we focus on the prefilling stage and propose a novel token pruning method named FTP for long-context LLM inference. Our approach is based on the observation that the FFN module accounts for over 60% of the inference time. FTP reduces this by pruning non-critical tokens before the inference of FFN. The importance of each token, along with the quantity to be pruned, are dynamically determined by the attention scores in each layer. Unlike previous token pruning methods, FTP preserves a substantial amount of information of the pruned tokens through the residual connection, thereby achieving a notable speedup with only a negligible decrease in performance. Specifically, the Qwen2-7B-Instruct model with FTP achieves a speedup of $1.24\times$ in the prefilling stage with only a 1.30% performance drop compared to the baseline model. The speedup is further boosted to $1.39\times$ on a Qwen1.5-32B-Chat model. Extensive experiments on long-context datasets across various tasks demonstrate the potential and effectiveness of FTP.

## 1 Introduction

Large language models (LLMs) have shown remarkable performance in a wide range of natural language processing applications, including summarization, question-and-answer, dialogue systems, and contextual learning (Thoppilan et al., 2022; Yuan et al., 2022; Wei et al., 2022; Zhang et al., 2023a). Recently, long-context models have emerged as an important trend in the development of LLMs, and several of them have already been launched to process complex and long prompts. For example, GPT4 (Achiam et al., 2023) and Qwen2 (Yang et al., 2024a) are both capable of $128k$ context length, and Claude-3 (Anthropic, 2024) can even process up to $200k$ tokens.

Given the advantages of long-context models, they nevertheless pose enormous challenges at inference time. As depicted in Figure 1, the inference of transformer-based LLMs with a KV cache typically consists of two stages: 1) the prefilling stage, which applies the self-attention, normalization, and the Feed-Forward Network (FFN) modules to all input token parallelly, records the key and value in the KV cache, and finally produces the first generated token. 2) the decoding stage, which takes the last generated token as input and iteratively predicts the next token with the input and the KV cache. We refer to the time spent in the prefilling stage as *time to first token (TTFT)*.

While there have been various methods trying to reduce the inference time of long-context LLMs by pruning the input prompt (Jiang et al., 2023b;a; Pan et al., 2024), compressing the KV cache (Li et al., 2024; Zandieh et al., 2024), improving the efficiency of decoding (Leviathan et al., 2022; Sun et al., 2023; Chen et al., 2023), and optimizing the attention kernel with sparse computation (Jiang et al., 2024), few works (Yang et al., 2024b; Fu et al., 2024) place attention to the reduction of TTFT. However, the computationally intensive nature of the prefilling stage implies that the TTFT for long-

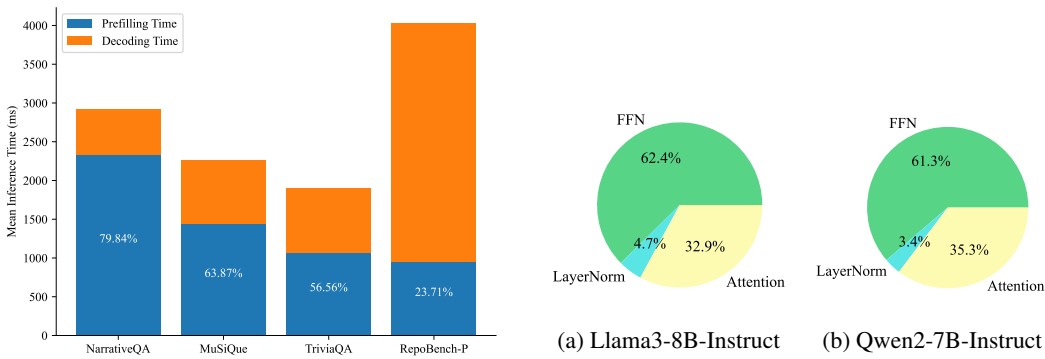

Figure 1: An illustration of the typical inference process of a transformer-based LLM with a KV cache. The first stage is called *prefilling*, which processes all input tokens in parallel and yields the first output token. The second stage is called *decoding*, which iteratively takes the latest generated token as input and generates the subsequent token until the termination condition is satisfied.

Figure 2: Prefilling proportion during inference. The average prompt lengths are 18409, 11214, 8209, and 4206 respectively. The average generating lengths are 12.08, 16.91, 17.10, and 61.10 respectively.

Figure 3: Walltime proportion of the main modules in each decoder layer during prefilling. The time proportion is averaged across all test samples in the TriviaQA dataset.

context inputs is nontrivial. We profile the prefilling and decoding time with the Qwen2-7B-Instruct model on the NarrativeQA (Kočiský et al., 2018), MuSiQue (Trivedi et al., 2022), TriviaQA (Joshi et al., 2017), and the RepoBench-P (Liu et al., 2023) datasets in LongBench (Bai et al., 2023b). For example, in Figure 2, the TTFT accounts for up to 80% of inference time on the NarrativeQA dataset, representing an obstacle to the application of LLMs in long-context scenarios.

To tackle this problem, we focus on the TTFT optimization of long-context LLM inference and propose a novel token pruning method named FFN Token Pruning (FTP) to alleviate computational demands in the prefilling stage. We first investigate the time proportion of the main modules (*i.e.*, the self-attention, normalization, and FFN module) in each layer by conducting experiments on the TriviaQA (Joshi et al., 2017) dataset and profiling the walltime spent on each module. As shown in Figure 3, the FFN module consistently takes a large proportion of inference time on both the Llama3 (AI@Meta, 2024) model and the Qwen2 (Yang et al., 2024a) model, suggesting a large potential for TTFT acceleration by reducing the tokens involving the FFN computation.

Unlike previous token pruning methods (Xiao et al., 2024; Li et al., 2024; Zhang et al., 2023b; Yang et al., 2024b) where tokens are directly pruned from the whole layer, FTP dynamically selects a certain proportion of tokens with an attention-based strategy and prunes them before the inference of FFN. For the pruned tokens, their FFN outputs are logically set to zeros vectors and they remain identical before and after the FFN due to the residual connection. Since the attention scores typically concentrate on a small proportion of tokens, a substantial amount of the FFN calculation can be circumvented, thereby reducing the TTFT. Furthermore, since the attention mechanism of each layer is fully preserved, the underlying context information associated with the tokens is substantially retained, leading to negligible accuracy loss. Extensive experiments on LongBench (Bai et al., 2023b) have demonstrated that our method has a balanced performance both in speedup and accuracy.

## 2 RELATED WORK

LLM inference optimization for long-context inputs has emerged as a significant area of NLP research. Several prior works (Dao et al., 2022; Kwon et al., 2023; Jiang et al., 2024) directly optimize the computing and memory footprint underlying the common operations in LLMs. In addition to these approaches, a substantial body of work concentrate on the logical inference of the model and optimize the inference process through two kinds of strategies: token pruning and layer skipping.

### 2.1 TOKEN PRUNING

Token pruning for sequence models has become a popular resesarch field before the era of LLM. Ren et al. (2023) proposes a model architecture named SeqBoat to skip non-activated sub-modules. Kim et al. (2022) and Guan et al. (2022) learn certain thresholds or parameterized modules to facilitate token pruning. For LLMs, many existing works (Liu et al., 2024; Adnan et al., 2024; Nawrot et al., 2024; Feng et al., 2024; Dong et al., 2024) accerlate inference by dropping irrelevant tokens, based on the observation that the attention scores are sparse in most layers. StreamingLLM (Xiao et al., 2024) only preserves the attention sink (*i.e.*, initial tokens) as well as the recent tokens to enable efficient infinite prompt requests. H2O (Zhang et al., 2023b) introduces the concept of "heavy hitter" and evicts tokens according to the cumulative attention score. SnapKV (Li et al., 2024) votes important previous tokens with the attention scores obtained by the "observation window", and prunes the unimportant ones to reduce the KV cache used in decoding. While these studies effectively compress the computation and memory footprint, they mainly accelerate the decoding phase without reducing the TTFT. On the other hand, PyramidInfer (Yang et al., 2024b) prunes tokens in both the prefilling and decoding phase, only reserving the "pivotal context" tokens in the KV cache. LazyLLM (Fu et al., 2024) also drops tokens from the prefilling stage and proposes an aux cache to avoid redundant computing, enabling the model to pick any subset of the input tokens at each layer, even if some tokens are already pruned. However, these methods either yield subtle speedup during prefilling or defer some computation to the decoding stage. More importantly, they prioritize the attention module and overlook the significant computation demands in the FFN module in practice, where flash attention (Dao et al., 2022) is a prevalent choice for efficient attention computation. Our work accelerates prefilling by pruning non-critical tokens prior to the FFN inference.

### 2.2 LAYER SKIPPING

Layer skipping accelerates the LLM inference by skipping the computation of some decoder layers. LayerSkip (Elhoushi et al., 2024) proposes a layer dropout strategy and an early exit loss during training, enabling layer skipping during inference, without a significant accuracy drop. MoD (Raposo et al., 2024) learns a router to determine for a token whether to skip both the self-attention and MLP blocks in a layer. Tyukin et al. (2024) investigates three types of layer skipping: 1) skipping MLP layers, 2) skipping attention layers, and 3) skipping transformer blocks on the 7B and 13B Llama2 Touvron et al. (2023) models, and observes that dropping the attention modules leads to much lower decrease in performance than dropping the FFN modules. Although our work focuses on reducing the computation in the FFN modules, the modules are not entirely bypassed, which results in slight degradation in performance.

## 3 FTP

### 3.1 LLM INFERENCE PERFORMANCE ANALYSIS

The inference of common LLMs comprises two phases: *prefilling* and *decoding*. The prefilling phase processes all prompt tokens in parallel, involving numerous GEMM operations, while the decoding phase iteratively takes the last generated token as input and predicts the next token until the stop condition is met.

As most existing LLMs utilize the transformer architecture, their inference process usually involves the forward of the stack of decoder layers, each of which mainly consists of three modules: normalization, self-attention and feed-forward network (FFN). As shown in Figure 3, the FFN module takes up over 60% of walltime of each decoder layer, highlighting a clear opportunity for accel-

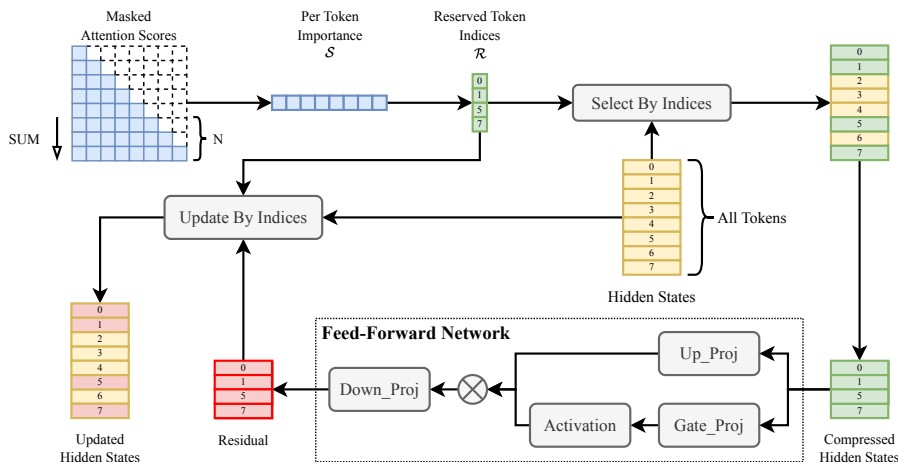

Figure 4: Overview of FTP in a layer. FTP starts from the end of the self-attention module. First, FTP obtains the attention scores from the last $N$ queries to all keys, and sums them in the *head* and *query* dimension (note that the *head* dimension is omitted in the figure) to derive the importance score $\mathcal{S}$ for each token. Next, the indices $\mathcal{R}$ of the tokens to prune are determined via Equation 2. Then token selection is performed based on $\mathcal{R}$, and only the chosen tokens are processed through the FFN. Finally, the FFN results (*i.e.*, residual) are utilized to update the corresponding token as specified by $\mathcal{R}$, while the unimportant tokens are not updated during the FFN inference in this layer.

eration. Taking two commonly encountered models, Llama3 (AI@Meta, 2024) and Qwen2 (Yang et al., 2024a), as examples, the FFN module takes the features of a sequence of $L$ tokens $\mathcal{T} \in \mathbb{R}^{L \times C}$ as input, and processes them token-wisely according to the function below:

$$FFN(\mathcal{T}) = (\mathcal{T}'W_U \odot A(\mathcal{T}'W_G))W_D, \tag{1}$$

where $\mathcal{T}' = Norm(\mathcal{T})$ are the normalized features, $A(\cdot)$ refers to the activation fucntion (*e.g.*, SiLU (Elfwing et al., 2018) in Llama3), and $W_D \in \mathbb{R}^{I \times C}, W_U \in \mathbb{R}^{C \times I}, W_G \in \mathbb{R}^{C \times I}$ refers to the *Down Projection*, *Up Projection* and *Gate Projection* matrices respectively. The formula indicates that most operations of FFN take place in the large and dense matrix multiplication with the three weight matrices, and the total FLOPs of these three matrix multiplication sum up to $6LCI$ (*i.e.*, $2LCI$ for one matrix multiplication). Note that other LLMs probably have slightly different FFN architectures, but the most computationally expensive part of theirs are still the matrix multiplication, which matches our analysis. Since both the hidden size of tokens (*i.e.*, $C$) and the size of weight matrices (*i.e.*, $C$ and $I$) are fixed in an off-the-shelf model, reducing the sequence length $L$ emerges as a viable strategy for FFN acceleration.

## 3.2 TOKEN PRUNING FOR FFN

Based on the analysis above, in this work, we propose to reduce the number of tokens before feeding them into FFN to accelerate the prefilling stage. Let us denote the "important" and "unimportant" sets of tokens as $\mathcal{T}_I \in \mathbb{R}^{L_I \times C}$ and $\mathcal{T}_U \in \mathbb{R}^{L_U \times C}$, where $L_I$ and $L_U$ are the number of "important" and "unimportant" tokens. As shown in Figure 4, only the "important" tokens are involved in the FFN calculation as formulated in Equation 1, and the FFN results for the "unimportant" tokens are logically set to zero vectors (*i.e.*, $FFN(\mathcal{T}_U) = \mathbf{0}$). Thanks to the residual connection surrounding the FFN module, setting the FFN results to be zeros vectors is equivalent to bypassing the updates in the FFN. Consequently, the FLOPs required for the FFN are reduced to $6L_ICI$ from $6LCI$.

To this end, two critical questions arise: 1) how to divide the input tokens into "important" and "unimportant" ones, and 2) how many tokens could be pruned, without causing obvious harm to the model performance? We analyse and discuss them in the following Section 3.2.1 and Section 3.2.2.

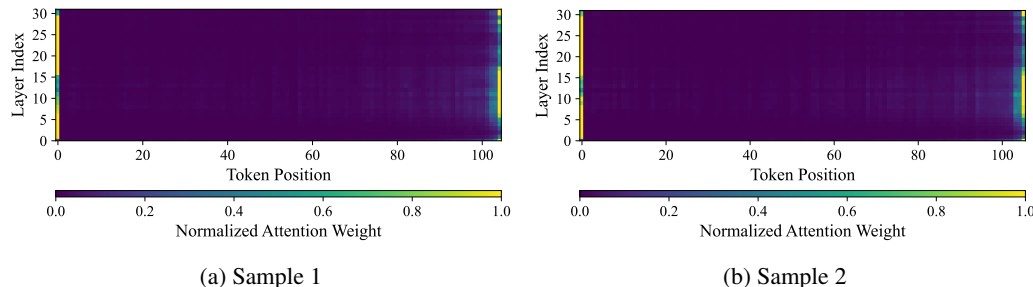

(a) Sample 1          (b) Sample 2

Figure 5: Visualization of the normalized attention weight for each layer. The experiment is conducted on a Llama3-8B-Instruct model on the Qasper dataset. Since the input length is too long to visualize, we divide the tokens into several groups with a group size of 32. Each column in the graph represents the average attention weight of a group of tokens across all layers. More visualization of the samples is in Appendix 6.2.

### 3.2.1 TOKEN PRUNING WITH ATTENTION SCORE

Token pruning is a common method to compress the input to LLM modules. Many existing works (Li et al., 2024; Fu et al., 2024; Yang et al., 2024b) propose to evaluate the importance of tokens with either the exact attention score matrix (*i.e.*, $softmax(\frac{QK^T}{\sqrt{d_k}})$) or an estimated one. Inspired by these studies and the computational order of self-attention and the FFN module, we propose to evaluate the importance of the tokens with the attention score from the current layer.

To investigate whether the attention score is suitable for token pruning for FFN, we visualize the attention score of layers from different depths in the Llama3-8B-Instruct (AI@Meta, 2024) model. As shown in Figure 5, we discover that 1) the attention scores in each layer are typically concentrated on a small number of tokens; 2) tokens with high attention scores in one layer may not necessarily be prioritized in the same manner across other layers; 3) the number of tokens with high attention scores varies among different layers.

Based on these observations, we propose an attention-based approach to evaluate the importance of the tokens in each layer. Let us denote the attention score matrices for all attention heads as $\mathcal{M} = \{M_h \in \mathbb{R}^{L \times L}\}_{h=1}^{H} = \{softmax(\frac{Q_h K_h^T}{\sqrt{d_k}})\}_{h=1}^{H}$, where $H$ is the number of attention heads. Since prior work (Li et al., 2024) has empirically revealed that the attention pattern obtained by the queries at the end of the prompts is nearly consistent with that obtained by all queries, we only retain the attention scores $\mathcal{M}'$ from the last $N$ queries to reduce overhead (*i.e.*, $\mathcal{M}' = \{M_h' \in \mathbb{R}^{N \times L}\}_{h=1}^{H}$). The attention scores $\mathcal{M}'$ are summed over all heads and queries to form the importance score $\mathcal{S}$ for each token. Inspired by Xiao et al. (2024), we statically reserve the initial $P$ and the last $N$ tokens in the prompts to maintain accuracy. These tokens are excluded from $\mathcal{S}$, resulting in the length of $\mathcal{S}$ being $L - P - N$. After that, the reserved token indices $\mathcal{R}$ can be calculated as

$$\mathcal{R} = \{i | i \in \mathbb{Z} \wedge 1 <= i <= P\}$$
$$\cup \{i + P | i \in \Gamma(\mathcal{S}, k)\}$$
$$\cup \{i | i \in \mathbb{Z} \wedge L - N < i <= L\}, \tag{2}$$

$$k = \min_{\Sigma_{i \in \Gamma(S,n)} S_i >= \eta \Sigma_i S_i} n, \tag{3}$$

where $\Gamma(A, a)$ returns the set of indices corresponding to the top $a$ elements in vector $A$. In this way, we reserve the top $k$ tokens such that their proportions of importance scores sums up to the reserve ratio $\eta$, as well as the initial $P$ and last $N$ tokens. The pseudo-code for FTP is in Algorithm 1.

Given that the attention scores tend to concentrate on a small number of tokens, a substantial number of tokens can be pruned in each layer even when $\eta$ approaches to 1.0. More importantly, the importance score $\mathcal{S}$ is computed with little overhead at each layer, which alleviates the problem reflected by the second observation. Since we use a reserved ratio $\eta$ rather than a static number to determine the number of tokens to retain, our approach is capable of handling various distributions of attention scores in different layers, as revealed in the third observation.

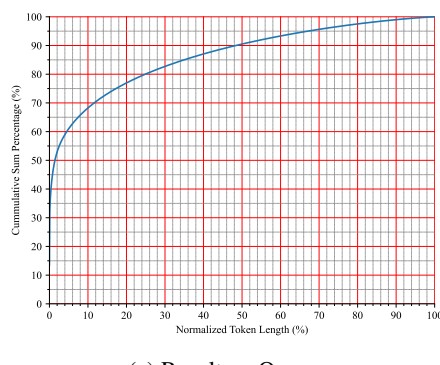

(a) Result on Qasper

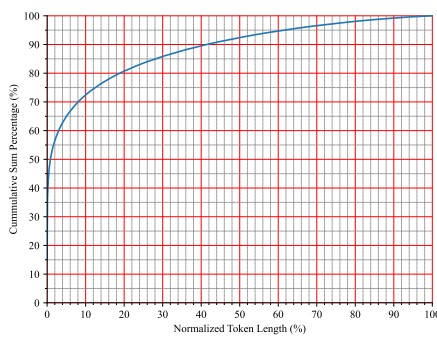

(b) Result on HotpotQA

Figure 6: The cumulative percentage of attention scores *w.r.t.* the normalized token length. The experiments are conducted on the Qasper (Dasigi et al., 2021) and HotpotQA (Yang et al., 2018) dataset with a Llama3-8B model. The results are averaged among all decoder layers and all samples.

### 3.2.2 TOKEN PRUNING PROPORTION

Since the number of pruned tokens is determined by the reserved ratio $\eta$ in each layer, we quantitatively investigate the relationship between $\eta$ and the number of reserved tokens by conducting an experiment on the Qasper (Dasigi et al., 2021) and HotpotQA (Yang et al., 2018) dataset with a Llama3-8B (AI@Meta, 2024) model. Concretely, we first derive the attention score matrices at each layer, average the scores along the *head* and *query* dimension, and obtain the averaged attention score for each token at each layer. Next, for each layer, we sort the tokens descendingly by their averaged attention score. Then we uniformly divide the input length into 100 steps, and record the cumulative percentage of attention score at each step. Finally, we average the cumulative percentage over all layers and all samples in the dataset, and the result is depicted in Figure 6. As shown in the result, a large proportion of attention score (*i.e.*, 95%) is occupied by only 60% of tokens, which is consistent with the first observation in Section 3.2.1, and suggests an opportunity for token pruning.

On the other hand, we empirically discover that shallow layers are more sensitive to our FTP (see Section 4.6), which is consistent with the conclusions of previous works (Fu et al., 2024; Yang et al., 2024b). However, we propose to preserve the whole layer (*i.e.*, $\eta = 1.0$) for the first $\mathcal{F}$ layers, and apply token pruning to the following layers. Thanks to the high pruning efficiency of FTP, the model maintains a considerable overall pruning rate.

---

**Algorithm 1** PyTorch Pseudo Code for FTP

---

**Input:** current layer index $c$, hidden state $X$, number of attention heads $H$, $\eta$, $\mathcal{F}$, $L$, $P$ and $N$
 1: $\mathcal{O}, \mathcal{M}$ = self-attention($X$)
 2: **if** $c > \mathcal{F}$ **then**
 3:     $\mathcal{S} = \mathcal{M}$.sum(0).sum(0)[$P{:}{-}N$]  # Assume $\mathcal{M}$ has a size of $(H, L, L)$
 4:     score, idx = sort($\mathcal{S}$, descending=True)
 5:     cumSum = cumsum(score)
 6:     threshold = cumSum[$-1$] $*\eta$
 7:     n = where(cumSum > threshold)[0] $+1$  # The first index that meets the condition
 8:     idx = idx[:n] + P  # Select the top n indices and shift them by P
 9:     $\mathcal{R}$ = cat(arange(P), idx, arange($L - N, L$))
10:     residual = FFN($\mathcal{O}[\mathcal{R}, :]$)
11:     $\mathcal{O}[\mathcal{R}, :]$ += residual
12: **else**
13:     residual = FFN($\mathcal{O}$)
14:     $\mathcal{O}$ += residual
15: **end if**
**Output:** Output hidden state $\mathcal{O}$

---

| Tasks | Method | Llama3-8B-Instruct | | Qwen2-7B-Instruct | |
|---|---|---|---|---|---|
| | | Score | TTFT Speedup ($\times$) | Score | TTFT Speedup ($\times$) |
| Single-Document QA | Baseline | **37.20** | 1.00 | **39.00** | 1.00 |
| | LLMLingua2 | 29.29 | 0.71 | 30.24 | 0.98 |
| | PyramidInfer* | 31.33 | 0.56 | / | / |
| | PyramidInfer | 32.05 | 1.02 | 29.19 | 1.21 |
| | Ours | 36.06 | **1.20** | 38.75 | **1.22** |
| Multi-Document QA | Baseline | **36.85** | 1.00 | **37.48** | 1.00 |
| | LLMLingua2 | 30.43 | 0.72 | 31.38 | 1.10 |
| | PyramidInfer* | 33.92 | 0.53 | / | / |
| | PyramidInfer | 33.43 | 0.97 | 32.86 | 1.19 |
| | Ours | 34.85 | **1.21** | 35.21 | **1.26** |
| Summarization | Baseline | **26.80** | 1.00 | **26.70** | 1.00 |
| | LLMLingua2 | 23.83 | 0.73 | 23.54 | 1.01 |
| | PyramidInfer* | 24.20 | 0.62 | / | / |
| | PyramidInfer | 24.21 | 1.06 | 22.65 | 1.20 |
| | Ours | 24.41 | **1.19** | 25.01 | **1.23** |
| Few-shot Learning | Baseline | **69.33** | 1.00 | **70.17** | 1.00 |
| | LLMLingua2 | 38.73 | 0.82 | 42.88 | 1.08 |
| | PyramidInfer* | 66.69 | 0.53 | / | / |
| | PyramidInfer | 66.37 | 0.97 | 66.10 | 1.24 |
| | Ours | 67.55 | **1.21** | 69.11 | **1.25** |
| Synthetic | Baseline | **37.00** | 1.00 | **37.50** | 1.00 |
| | LLMLingua2 | 12.75 | 0.66 | 7.75 | 1.10 |
| | PyramidInfer* | 36.00 | 0.50 | / | / |
| | PyramidInfer | 35.50 | 0.94 | 35.50 | 1.16 |
| | Ours | 36.00 | **1.25** | 36.75 | **1.30** |
| Code Completion | Baseline | 55.17 | 1.00 | **58.43** | 1.00 |
| | LLMLingua2 | 31.47 | 0.69 | 37.69 | 0.88 |
| | PyramidInfer* | **55.29** | 0.66 | / | / |
| | PyramidInfer | 55.24 | 1.10 | 56.52 | **1.24** |
| | Ours | 35.91 | **1.19** | 56.74 | 1.22 |

Table 1: TTFT speedup and accuracy score across various long-context tasks on LongBench. The result for each task is computed by averaging the results of datasets belonging to the task. Note that the attention modules of all methods except for PyramidInfer* are implemented with flash attention, which is lossless in accuracy and efficient in time and memory, while PyramidInfer* is the official implementation with PyTorch-implemented attention. Additionally, PyramidInfer* encounters out-of-memory issues when applied to the Qwen2 model, which has a max context length of 32k.

## 4 EXPERIMENTS

**Datasets.** We conduct experiments on the LongBench (Bai et al., 2023b) benchmark, containing 16 long-context datasets for 6 types of tasks (*i.e.*, single-document QA, multi-document QA, summarization, few-shot learning, synthetic tasks, and code completion), which provides a comprehensive assessment on the long-context understanding capability of LLMs. The average context length across the datasets ranges from $5,000$ to $15,000$, with the number of test samples in each dataset typically being 200, except for the datasets of code completion, which contain 500 samples. We follow the official pipeline of LongBench to pre-process the data and evaluate the models.

**Metrics.** We utilize two kinds of metrics to assess the quality and efficiency of the generating process of the LLMs with our proposed FTP. The quality metric is the accuracy score from LongBench. Since the datasets in LongBench adopt various metrics for the accuracy evaluation, the accuracy score for each dataset is one of the following metrics: 1) F1 Score, 2) Rouge-L, 3) Accuracy, and 4) Edit Similarity. All of these metrics have a range from 0.0 to 1.0, hence we average them to represent the accuracy score of a task. The efficiency metric is the TTFT speedup (*i.e.* $\frac{TTFT_{baseline}}{TTFT_{FTP}}$) *w.r.t.* the baseline model (*i.e.*, inferencing the pre-trained model with the standard process).

### 4.1 Implementation Details

**Basic Configuration.** We implement FTP on the Llama3-8B-Instruct (AI@Meta, 2024), Qwen2-7B-Instruct (Yang et al., 2024a) and two larger models (see Section 4.5). The max context length for Qwen2 and Llama3 are $32k$ and $8k$ respectively, and the data samples that exceed the max context length of the models are truncated during the data pre-processing pipeline of LongBench. We set $P = 100$ and $N = 50$ for both models. For the Llama3-8B model, we set $\mathcal{F} = 10$ to preserve the first 10 layers and $\eta = 0.90$ for the following layers. When it comes to the Qwen2-7B model, $\mathcal{F}$ and $\eta$ are set as 10 and 0.95 respectively. Note that FTP is free of further training and shares the same weights with the original model. All experiments are conducted on NVIDIA A100 GPUs without model quantization.

**Attention Implementation.** Our method is implemented with the transformers library (Wolf et al., 2020) and flash attention, which is a prevalent choice in practical inference of LLMs. Given that flash attention inherently does not return attention weights, we recalculate the necessary attention weights to assess token importance, introducing only a negligible cost (refer to Section 4.6.1).

### 4.2 Results on LongBench

We conduct experiments on LongBench (Bai et al., 2023b), including six long-context tasks, to evaluate the accuracy score and TTFT speedup of our method. In addition to the baseline model, we incorporate a state-of-the-art prompt compression method (Pan et al., 2024) for comparison, which formulates prompt compression as a token classification problem and utilizes a transformer encoder with a linear classification head to prune the prompt before feeding them into the LLM.

The performance for each task is calculated by averaging the results of datasets associated with that specific task. As shown in Table 1, FTP generally presents considerable speedup across all six long-context tasks with a negligible drop in accuracy score. Thanks to the longer context support (*i.e.*, $32k$) of Qwen2-7B-Instruct, FTP achieves even greater speed improvements, despite applying a higher reserved ratio $\eta$. Furthermore, LLMLingua2 (Pan et al., 2024) on both models can hardly accelerate the pre-filling stage even with a compression ratio of 0.2.

### 4.3 Comparison with State-of-The-Art Method

We conduct a comparative analysis between FTP and PyramidInfer (Yang et al., 2024b), a state-of-the-art method of prefilling acceleration. PyramidInfer accelerates both the prefilling and decoding stages by compressing the KV cache with different compression rates for each layer.

We conduct experiments on the Llama3-8B-Instruct and Qwen2-7B-Instruct models, and the results are in Table 1. We present two versions of PyramidInfer for comparison. The official implementation of PyramidInfer (denoted as PyramidInfer* in Table 1) utilizes PyTorch-implemented attention module and fails to accelerate the prefilling stage. Furthermore, it encounters out-of-memory issues when applied to the Qwen2 model, which has a max context length of 32k. On the other hand, we re-implement PyramidInfer (denoted as PyramidInfer in Table 1)with flash attention and re-calculate the necessary attention weights (*i.e.*, 20% attention weights following the official setting) for it. At this time, PyramidInfer gets rid of out-of-memory issues on Qwen2 model but is still slower with more accuracy degradation than FTP in most tasks.

### 4.4 Accuracy vs. TTFT Speedup

The TTFT speedup of FTP can be modulated from two aspects: 1) the number of layers to which FTP is applied, and 2) the reserved ratio $\eta$ at each decoder layer. We further examine the variation of accuracy score *w.r.t.* TTFT speedup by evaluating the accuracy score at various levels of inference efficiency. We incorporate LLMLinuga-2 for comparison and adjust the prompt compression rate to achieve different speedups. As shown in Figure 7, FTP achieves notable TTFT speedup with negligible performance drop across various tasks, presenting an effective balance between speedup and accuracy. Moreover, the accuracy score of FTP even surpasses that of the baseline in certain tasks (*e.g.*, Single-Document QA and Synthetic Task). Conversely, LLMLingua2 experiences an evident decline in accuracy to accelerate the prefilling stage.

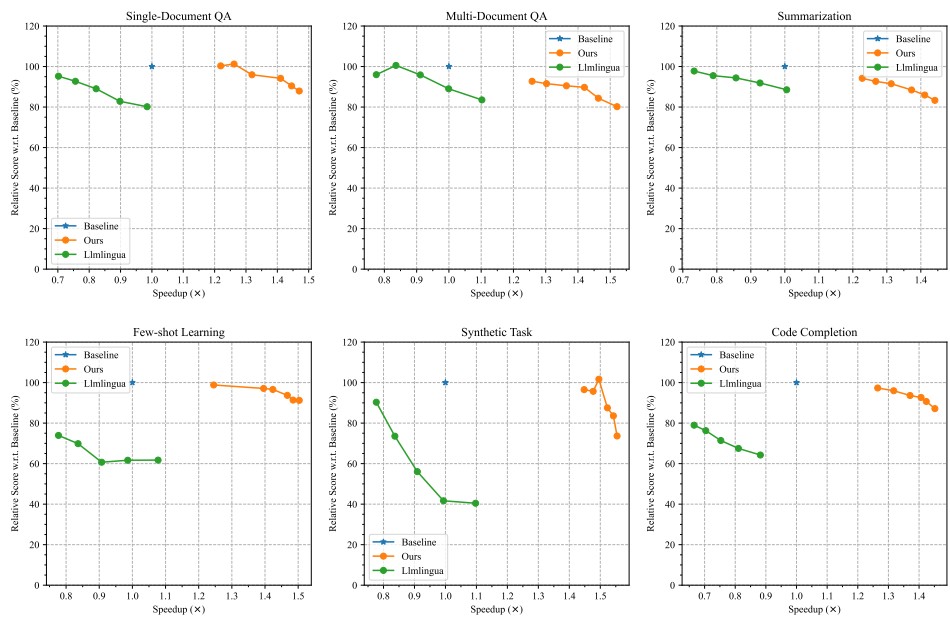

Figure 7: TTFT speedup *vs.* accuracy comparison for Qwen2-7B-Instruct across different tasks.

## 4.5 FTP ON LARGER MODELS

In this section, we further investigate the effectiveness of FTP on larger LLMs and evaluate our method on Qwen1.5-32B-Chat (Bai et al., 2023a) and Qwen2-72B-Instruct (Yang et al., 2024a), both of which support a max context length of 32k. For the Qwen1.5-32B-Chat model, we preserve the first $\mathcal{F} = 10$ layers and set $\eta = 0.90$ for the subsequent 54 layers. For the Qwen2-72B-Instruct model, we preserve the first $\mathcal{F} = 10$ and set $\eta = 0.93$ for the following 70 layers. As shown in Table 2, FTP facilitates a significant speedup across all tasks with a subtle impact on the accuracy score, which indicates the potential of practical application of FTP.

We attribute this enhanced acceleration on the larger models to two primary factors. Firstly, larger models have deeper architectures. Therefore, compared to the smaller models, more tokens can be pruned even though the first $\mathcal{F}$ layers are fully preserved. Secondly, these models have up to $4\times$ and even $10\times$ of weights, and they exhibit robustness on accuracy even when a substantial number of tokens are pruned.

| Tasks | Method | Qwen1.5-32B-Chat | | Qwen2-72B-Instruct | |
|---|---|---|---|---|---|
| | | Score | TTFT Speedup ($\times$) | Score | TTFT Speedup ($\times$) |
| Single-Document QA | Baseline | **40.68** | 1.00 | **43.78** | 1.00 |
| | Ours | 37.16 | **1.37** | 41.36 | **1.31** |
| Multi-Document QA | Baseline | **44.98** | 1.00 | **62.70** | 1.00 |
| | Ours | 42.06 | **1.39** | 60.92 | **1.33** |
| Summarization | Baseline | **25.91** | 1.00 | **28.32** | 1.00 |
| | Ours | 23.78 | **1.37** | 26.80 | **1.32** |
| Few-shot Learning | Baseline | 66.99 | 1.00 | **69.87** | 1.00 |
| | Ours | **67.74** | **1.40** | 67.15 | **1.33** |
| Synthetic | Baseline | **52.67** | 1.00 | **54.50** | 1.00 |
| | Ours | 46.25 | **1.45** | 51.00 | **1.34** |
| Code Completion | Baseline | **46.97** | 1.00 | **69.05** | 1.00 |
| | Ours | 46.63 | **1.39** | 68.35 | **1.36** |

Table 2: TTFT speedup and accuracy on various tasks on Qwen1.5-32B and Qwen2-72B models.

### 4.6 ABLATION STUDY

We conduct an ablation study of FTP on the approaches and hyper-parameters we proposed in Section 3.2. The experiment for our token pruning strategy is in Section 4.6.1, and more analysis can be found in Appendix 6.1.

#### 4.6.1 TOKEN PRUNING STRATEGY

To evaluate the token pruning strategy proposed in Section 3.2.1, we replace our attention-based strategy with a random selecting strategy and evaluate the model performance with Llama3-8B-Instruct and Qwen2-7B-Instruct. Concretely, we first record the number of pruned tokens in each layer for each sample in our approach, and then randomly prune the same number of tokens in the "random" variant. As illustrated in Table 3, our attention-based strategy is crucial in determining which tokens to prune. The model accuracy consistently suffers a significant drop across all tasks when applied a random pruning strategy. On the other hand, the TTFT of FTP is only marginally higher than that of the random variant, indicating that the computational cost introduced by FTP is trivial. Specifically, the computational cost introduced by FTP on Llama3 and Qwen2 models are 7-10ms and 8-15ms respectively, accounting for only 1%-3% and 0.8%-1.9% of the TTFT.

| Tasks | Method | Llama3-8B-Instruct | | Qwen2-7B-Instruct | |
|---|---|---|---|---|---|
| | | Score | TTFT (ms) | Score | TTFT (ms) |
| Single-Document QA | Baseline | **37.20** | 574.70 | **39.00** | 1122.30 |
| | Random | 11.14 | 469.99 | 20.63 | 907.75 |
| | Ours | 36.06 | 480.24 | 38.75 | 923.18 |
| Multi-Document QA | Baseline | **36.85** | 664.80 | **37.48** | 1080.73 |
| | Random | 7.56 | 541.11 | 9.82 | 845.63 |
| | Ours | 34.85 | 551.23 | 35.21 | 859.11 |
| Summarization | Baseline | **26.80** | 526.11 | **26.70** | 797.57 |
| | Random | 15.90 | 428.95 | 18.67 | 636.32 |
| | Ours | 24.41 | 438.08 | 25.01 | 647.59 |
| Few-shot Learning | Baseline | **69.33** | 594.20 | **70.17** | 810.22 |
| | Random | 21.58 | 481.82 | 33.81 | 630.81 |
| | Ours | 67.55 | 489.26 | 69.11 | 641.58 |
| Synthetic | Baseline | **37.00** | 732.98 | **37.50** | 1235.10 |
| | Random | 2.72 | 573.84 | 1.71 | 947.67 |
| | Ours | 36.00 | 584.18 | 36.75 | 955.94 |
| Code Completion | Baseline | **55.17** | 449.45 | **58.43** | 621.11 |
| | Random | 16.28 | 367.09 | 24.41 | 489.04 |
| | Ours | 35.91 | 375.22 | 56.74 | 498.15 |

Table 3: Accuracy and TTFT comparison for Llama3-8B-Instruct and Qwen2-7B-Instruct employing random pruning strategy and our attention-based strategy. The *random* variant prunes the identical number of tokens at each layer as ours.

## 5 CONCLUSION

In this work, we explore a novel approach of prefilling acceleration by pruning tokens for FFN, and propose the FFN Token Pruning (FTP) technique for long-context inference of LLMs, without additional training or finetuning. We start by analyzing the TTFT proportion in long-context inference of LLMs, and profiling the time proportion of the primary components in a typical LLM layer. Observing the substantial time allocation to FFN during prefilling, we reduces FFN computation time by "pruning" non-critical tokens with an attention-based strategy prior to FFN inference. In a addition, FTP preserves a large amount of information of the pruned toknes through the residual connection. Experiments on long-context datasets across various tasks indicate that FTP is capable of delivering significant acceleration while maintaining performance.

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

# 6 APPENDIX

## 6.1 ADDITIONAL ANALYSIS OF FTP

### 6.1.1 RESULTS ON L-EVAL

We conduct additional experients on the L-Eval (An et al., 2023) benchmark for a comprehensive evaluation. L-Eval has two groups of tasks: closed-ended tasks and open-ended tasks. The former group focus on evaluating an LLM's ability of reasoning and understanding a long context, while the latter one is more about summarization tasks for a long document. We run FTP and Pyramid-Infer (Yang et al., 2024b) on the Llama3-8B-Instruct and Qwen2-7B-Instruct models for both tasks, and the results are in Table 4 and Table 5 respectively.

For closed-ended tasks, FTP with the Llama3 model sustains considerable performance across half of the datasets, and achieves a TTFT speedup comparable to PyramidInfer. When applied on Qwen2-7B-Instruct model, FTP not only enhances the performance on the GSM (Cobbe et al., 2021) and CodeU datasets, but also surpasses the baseline scores. Furthermore, the longer context length for Qwen2 illustrates the acceleration of FTP during the prefilling stage, showcasing its efficiency.

For open-ended tasks, we employ GPT4 as a judge to compare the outputs from the model with those from GPT-3.5-turbo-16k-0613. Each pair of predictions is evaluated by GPT4 twice, with their positions swapped after the first judgement. The number of wins and ties for the model are recorded in Table 5. As illustrated in the results, FTP achieves higher TTFT speedup for both models, and notably, the performance on Qwen2-7B-Instruct even surpasses that of the baseline.

| Model | Coursera | GSM | QuALITY | CodeU | SFiction | TRL | TPO | Avg. | SpeedUp |
|---|---|---|---|---|---|---|---|---|---|
| Llama3 Baseline | 52.76 | **66.00** | **60.40** | **4.44** | **71.09** | **62.00** | 75.46 | **56.02** | 1.00 |
| Llama3 + FTP | **53.20** | 38.00 | 55.45 | 4.44 | 69.53 | 52.00 | 74.35 | 49.57 | 1.21 |
| Llama3 + PyramidInfer | 52.18 | 68.00 | 60.40 | 4.44 | 71.09 | 49.33 | **75.84** | 54.47 | **1.24** |
| Qwen2 Baseline | 68.60 | 24.00 | **65.35** | 7.78 | **73.44** | 47.33 | **81.04** | 52.51 | 1.00 |
| Qwen2 + FTP | 68.60 | **30.00** | 65.35 | **10.00** | 72.66 | 43.33 | 80.30 | **52.89** | 2.50 |
| Qwen2 + PyramidInfer | **69.77** | 30.00 | 65.35 | 6.67 | 71.88 | 42.67 | 81.04 | 52.48 | 2.01 |

Table 4: L-Eval results on closed-ended tasks. "TRL" stands for the "topic retrieval longchat" dataset, and "SpeedUp" represents the TTFT speedup *w.r.t.* the baseline.

| Model | Wins | Ties | Win-rate (%) | TTFT SpeedUp |
|---|---|---|---|---|
| Llama3 Baseline | 50 | 62 | **42.19** | 1.00 |
| Llama3 + FTP | 44 | 67 | 40.36 | **1.19** |
| Llama3 + PyramidInfer | 49 | 51 | 38.80 | 1.08 |
| Qwen2 Baseline | 42 | 75 | 42.97 | 1.00 |
| Qwen2 + FTP | 47 | 69 | **43.58** | **1.21** |
| Qwen2 + PyramidInfer | 35 | 69 | 37.37 | 1.09 |

Table 5: L-Eval results on open-ended tasks. The output of models are compared with that of GPT-3.5-turbo-16k-0613 by the judge model (*i.e.*, GPT4).

### 6.1.2 NEEDLE-IN-A-HAYSTACK EVALUATION

In this section, we evaluate whether FTP results in the loss of intermediate information by conducting the Needle-in-a-Haystack (Kamradt, 2023) experiment. The Needle-in-a-Haystack test inserts a specific statement (referred to as the "needle") in the middle of a long-context document (the "haystack"), and requires the LLM to retrieve this statement. For evaluation, a GPT4 (Achiam et al., 2023) model is utilized to score the relevance between the LLM's output and the "needle".

We iterate the context length from 1000 to the max context length of the model, and position the "needle" from the beginning (0.0%) to the end (100%) in steps of 10% for each context length. The result for the Llama3-8B-Instruct and Qwen2-7B-Instruct models are presented in Figure 8. As demonstrated in the results, the Qwen2 baseline model exhibits substantial retrieval capabilities. Our method appears to have minimal impact on this feature, successfully retrieve the "needle" across nearly all context lengths and depths of the "needle". For the Llama3 model, FTP even enhances performance as the context length approaches its limit. The results in Figure 8 suggest that FTP does not lead to a loss, and may even reduce the loss of intermediate information.

### 6.1.3 THE IMPACT OF INITIAL AND THE LAST TOKENS

In this section, we tune the hyper-parameters $P$ and $N$ mentioned in Section 3.2.1 to analyse their impact on model accuracy. The experiments are conducted on the Qasper (Dasigi et al., 2021) HotpotQA (Yang et al., 2018) dataset, with a Llama3-8B-Instruct model, $\mathcal{F} = 10$ and $\eta = 0.95$, and the results are in Table 6. The results demonstrates that choosing an excessively small $P$ or $N$ yields significant decrease in accuracy, although they reduce the computational overhead of FTP. Nonetheless, an increase in $P$ or $N$ does not necessarily translate to improved accuracy. Moreover, a larger $N$ typically incurs additional computational cost and results in a reduced number of pruned tokens given the same $\eta$.

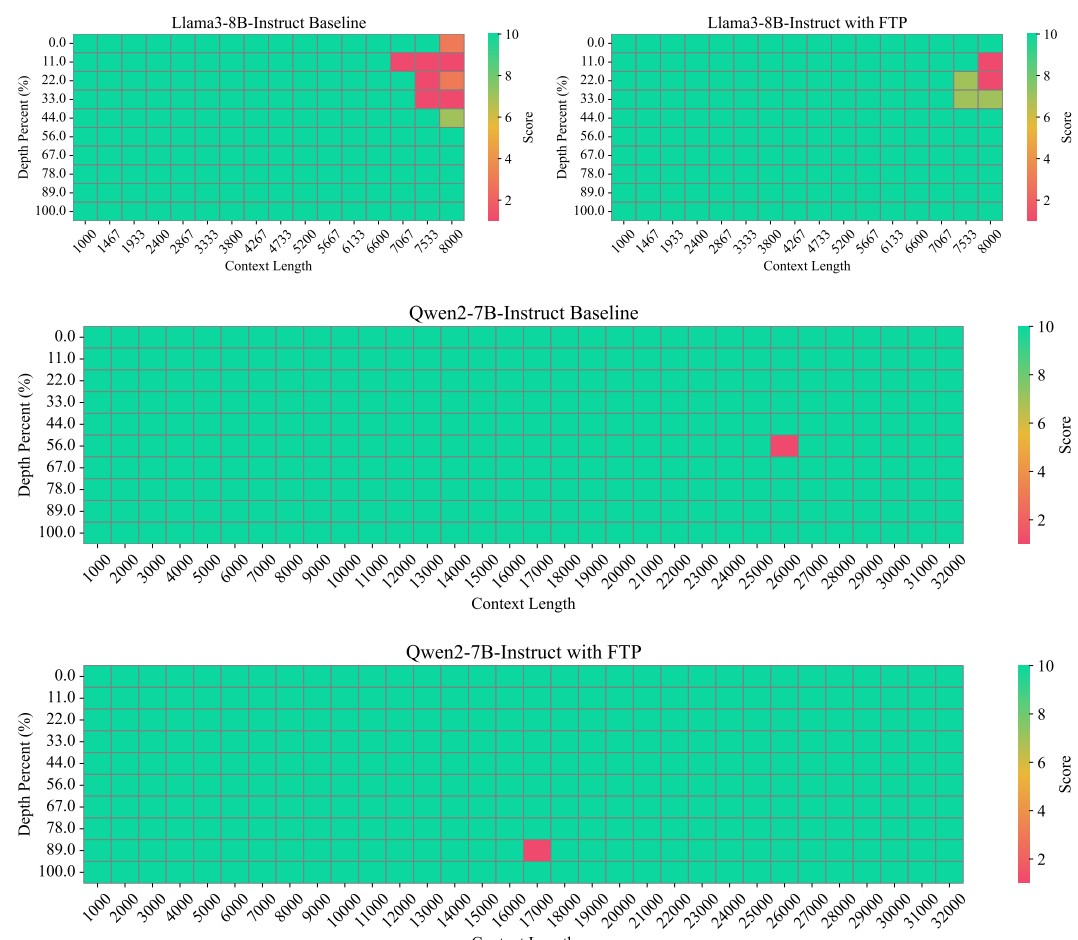

Figure 8: Visualization of the result of the Needle-in-a-Haystack test for Llama-3-8B-Instruct and Qwen2-7B-Instruct with and without FTP. The x-axis represents the length of the documents and the y-axis denotes the position of the statement "needle". The scores are given by GPT4 (Achiam et al., 2023) by evaluating the relevance of the model output and the "needle".

| N
P | 1 | 10 | 20 | 50 | 80 | 100 | 150 | 200 |
|---|---|---|---|---|---|---|---|---|
| 1 | 34.99 | 43.16 | 43.80 | 42.74 | 41.74 | 42.44 | 43.49 | 42.99 |
| 10 | 37.25 | 42.44 | 45.34 | 45.11 | 45.14 | 44.90 | 44.89 | 45.40 |
| 20 | 37.38 | 42.44 | 45.50 | 45.06 | 44.93 | 45.19 | 44.36 | 44.41 |
| 50 | 36.77 | 42.67 | 45.40 | 44.75 | 44.61 | 44.95 | 43.73 | 45.03 |
| 80 | 37.26 | 43.05 | 44.95 | 45.27 | 45.11 | 44.73 | 45.24 | 43.85 |
| 100 | 38.13 | 43.44 | 44.81 | 45.75 | 44.92 | 45.46 | 44.94 | 44.39 |
| 150 | 37.27 | 43.05 | 44.51 | 44.57 | 45.08 | 45.67 | 45.25 | 45.61 |
| 200 | 38.46 | 42.87 | 44.78 | 45.00 | 45.38 | 45.43 | 44.88 | 45.01 |

Table 6: Accuracy score under different settings of the initial and the last reserved tokens. The scores are averaged over Qasper and HotpotQA datasets.

### 6.1.4 RESERVED RATIOS

We also conduct experiments to analyse the impact of the reserved ratio $\eta$ for FTP as mentioned in Section 3.2.1. We statically reserve the first $\mathcal{F} = 10$ layers and set different reserved ratios $\eta$ for the subsequent layers of the Qwen2-7B-Instruct model. The experients are conducted on LongBench

| Tasks | $\eta = 0.65$ | $\eta = 0.70$ | $\eta = 0.75$ | $\eta = 0.80$ | $\eta = 0.85$ | $\eta = 0.90$ | $\eta = 0.95$ |
|---|---|---|---|---|---|---|---|
| Single-Document QA | 33.20 | 34.13 | 34.96 | 36.46 | 36.79 | 37.23 | 38.75 |
| Multi-Document QA | 29.69 | 30.28 | 30.53 | 32.24 | 34.23 | 34.60 | 35.21 |
| Summarization | 21.22 | 21.43 | 21.90 | 22.63 | 23.34 | 24.25 | 25.01 |
| Few-shot Learning | 64.13 | 64.29 | 65.17 | 66.08 | 68.20 | 67.34 | 69.11 |
| Synthetic | 25.00 | 29.00 | 31.75 | 33.25 | 35.75 | 34.50 | 36.75 |
| Code Completion | 50.22 | 50.92 | 52.35 | 54.14 | 53.97 | 56.08 | 56.74 |

Table 7: The accuracy score for each task in LongBench when applying different reserved ratios $\eta$ for FTP on the Qwen2-7B-Instruct model.

and the results of accuracy score and TTFT speedup are shown in Table 7 and Table 8 respectively. As shown in the results, FTP has a considerable TTFT speedup even when $\eta$ is set to $0.95$, suggesting a substantial number of tokens are pruned. More importantly, as $\eta$ is progressively reduced from $0.95$ to $0.65$, the speedup continues to increase, but the rate of increment in speedup gradually becomes smaller. This observation is consistent with our analysis in Section 3.2.2 and Figure 6. However, the decrease in accuracy does not follow this pattern. Therefore, FTP can achieve an optimal balance between accuracy and TTFT speedup when $\eta$ is approximately $0.80$.

| Tasks | $\eta = 0.65$ | $\eta = 0.70$ | $\eta = 0.75$ | $\eta = 0.80$ | $\eta = 0.85$ | $\eta = 0.90$ | $\eta = 0.95$ |
|---|---|---|---|---|---|---|---|
| Single-Document QA | 1.49 | 1.47 | 1.45 | 1.41 | 1.37 | 1.32 | 1.22 |
| Multi-Document QA | 1.54 | 1.52 | 1.49 | 1.46 | 1.42 | 1.36 | 1.26 |
| Summarization | 1.48 | 1.46 | 1.44 | 1.41 | 1.37 | 1.31 | 1.23 |
| Few-shot Learning | 1.50 | 1.48 | 1.46 | 1.43 | 1.40 | 1.35 | 1.25 |
| Synthetic | 1.55 | 1.54 | 1.52 | 1.50 | 1.46 | 1.40 | 1.30 |
| Code Completion | 1.48 | 1.45 | 1.44 | 1.41 | 1.37 | 1.32 | 1.22 |

Table 8: The TTFT speedup ($\times$) for each task in LongBench when applying different reserved ratio $\eta$ for FTP on the Qwen2-7B-Instruct model.

### 6.1.5 RESERVED LAYERS

In this section, we explore the impact of reserving shallow layers as mentioned in Section 3.2.2. We set different $\mathcal{F}$ to reserve different numbers of shallow layers and fix $\eta$ to $0.90$ for the following layers of Qwen2-7B-Instruct when applying FTP. The experiments are conducted on LongBench and the results of accuracy score and TTFT speedup are presented in Table 9 and Table 10 respectively. As shown in the results, the accuracy score increases as more shallow layers of the model are fully reserved. However, some certain tasks (*e.g.*, Few-Shot Learning and Synthetic) exhibit little changes in accuracy score or even demonstrate an increase in accuracy when $\mathcal{F}$ is incremented from 5 to 10. In general, the TTFT speedup increases as we decrease $\mathcal{F}$. However, there is an anomalous decline in speedup when $\mathcal{F}$ is reduced from 10 to 9. This anomaly could be explained by a change to the distribution of attention weights to a more uniform pattern at $\mathcal{F} = 9$ compared to $\mathcal{F} = 10$, resulting in a decrease in the number of pruned tokens.

| Tasks | $\mathcal{F} = 4$ | $\mathcal{F} = 5$ | $\mathcal{F} = 6$ | $\mathcal{F} = 7$ | $\mathcal{F} = 8$ | $\mathcal{F} = 9$ | $\mathcal{F} = 10$ |
|---|---|---|---|---|---|---|---|
| Single-Document QA | 33.70 | 33.99 | 36.00 | 35.73 | 36.56 | 36.81 | 37.23 |
| Multi-Document QA | 31.68 | 31.72 | 32.44 | 32.65 | 32.23 | 33.30 | 34.60 |
| Summarization | 22.24 | 22.61 | 22.69 | 22.95 | 23.38 | 23.78 | 24.25 |
| Few-shot Learning | 65.67 | 67.38 | 67.74 | 67.82 | 67.32 | 67.38 | 67.34 |
| Synthetic | 32.25 | 35.75 | 34.50 | 33.25 | 38.00 | 36.25 | 34.50 |
| Code Completion | 52.98 | 53.20 | 53.93 | 54.70 | 54.79 | 55.82 | 56.08 |

Table 9: The accuracy score for each task in LongBench when reserving different numbers of shallow layers for FTP on the Qwen2-7B-Instruct model.

| Tasks | $\mathcal{F} = 4$ | $\mathcal{F} = 5$ | $\mathcal{F} = 6$ | $\mathcal{F} = 7$ | $\mathcal{F} = 8$ | $\mathcal{F} = 9$ | $\mathcal{F} = 10$ |
|---|---|---|---|---|---|---|---|
| Single-Document QA | 1.42 | 1.40 | 1.39 | 1.37 | 1.35 | 1.30 | 1.32 |
| Multi-Document QA | 1.47 | 1.45 | 1.43 | 1.42 | 1.40 | 1.35 | 1.36 |
| Summarization | 1.42 | 1.41 | 1.39 | 1.37 | 1.36 | 1.30 | 1.31 |
| Few-shot Learning | 1.47 | 1.44 | 1.42 | 1.41 | 1.39 | 1.34 | 1.35 |
| Synthetic | 1.51 | 1.50 | 1.48 | 1.47 | 1.45 | 1.40 | 1.40 |
| Code Completion | 1.42 | 1.41 | 1.39 | 1.37 | 1.36 | 1.30 | 1.32 |

Table 10: The TTFT speedup ($\times$) for each task in LongBench when reserving different numbers of shallow layers for FTP on the Qwen2-7B-Instruct model.

## 6.2 ADDITIONAL VISUALIZATION OF ATTENTION WEIGHTS

To illustrate the generalizability of our findings discussed in Section 3.2.1, we visualize the attention weights of more samples from the Qasper (Dasigi et al., 2021) dataset, as presented in Figure 9.

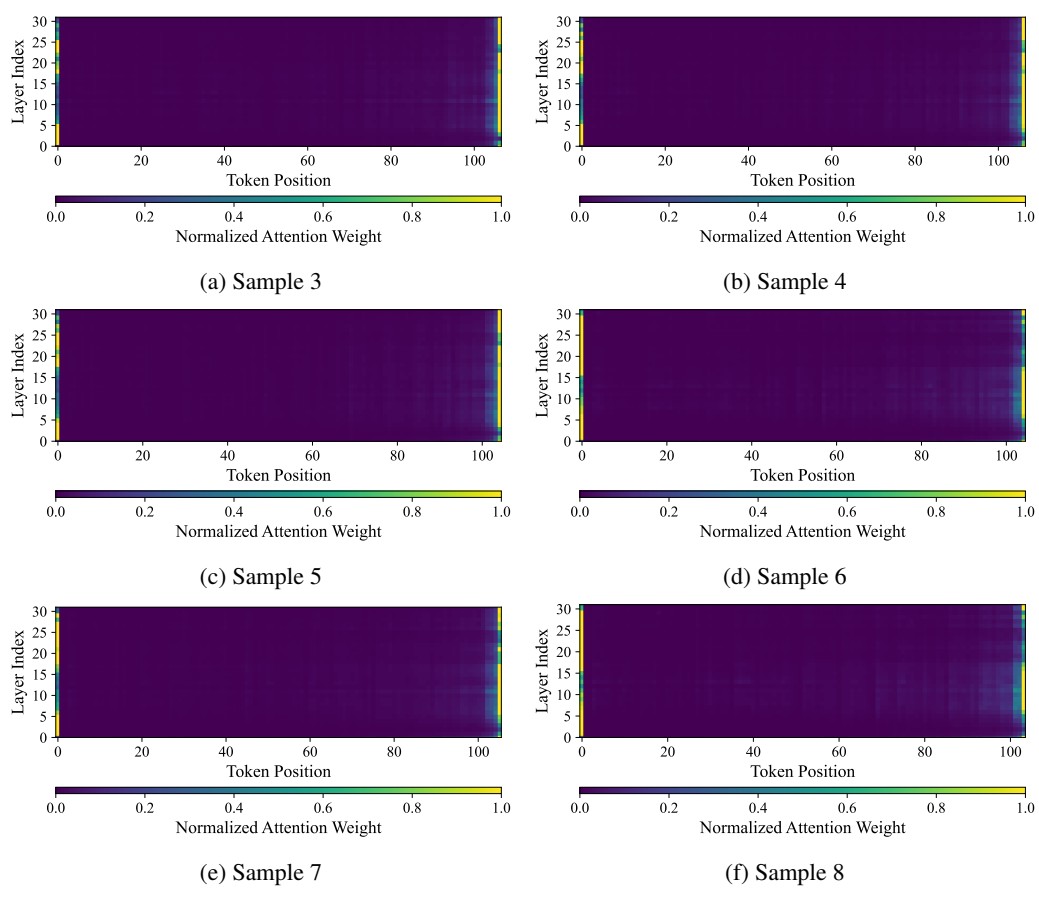

Figure 9: Additional visualization of the normalized attention weight for each layer. The experiment is conducted on a Llama3-8B-Instruct model on the Qasper dataset. Since the input length is too long to visualize, we divide the tokens into several groups with a group size of 32. Each column in the graph represents the average attention weight of a group of tokens across all layers.

