# OpenReview forum: "FTP: Efficient Prefilling for Long-Context LLM Inference via FFN Token Pruning"
_ICLR.cc/2025/Conference — Submitted to ICLR 2025_

### Official Review · Reviewer_Pbr1 · 2024-10-30

**Soundness:** 3
**Presentation:** 2
**Contribution:** 3
**Rating:** 6
**Confidence:** 3

**Summary:**

The paper introduces a new token pruning techniques at the pre-filling stage (before the autoregressive generative inference starts) for the FFN layer. This is mainly done by looking at the attention scores associated to each token, and keeping top-k tokens. The k is dynamically determined. Some heuristics are also employed  - for example, forcibly keeping first few and last few tokens from the original input and starting pruning only after the first few layers are executed. Performance in terms of accuracy and time taken till first token generation is shown in longBench.

**Strengths:**

1. The method is straightforward and elegant.
2. Offers a reasonable speedup while mostly maintaining the original performance.

**Weaknesses:**

1. No discussion about memory impact. Does this method reduces memory or keeps them the same? If there is a reduction - then by how much? Many of the prior methods are also motivated in terms of memory reduction.

1. Weak literature review. There are many works on token pruning outside LLM ([1,2,3] - just to name a few, but this is still a very incomplete list. More can be found by browsing the citation network). I found some of the contrasting against prior works as unsatisfactory. Overall, presentation can be more polished.

1. The overall trade-off (accuracy loss vs speedup) feels a bit lukewarm. The speedup is only (~1.2x-1.4x depending on model size), and in most contexts it may not seem worth it for the accuracy loss - although that's a matter of specific use case. While accuracy loss is bearable in most contexts, sometimes it can be high like about 20% drop by Llama for code-completion.

1. I am not sure how fair the comparison between PyramidInfer vs author's methods given the former doesn't have flash-attention and the latter do. Is PyramidInfer simply not implemented with flash-attention officially or can it be not implemented in principle? If it can be implemented in principle, then the only reason it would be currently facing a disadvantage seems to be due to lacking the relevant implementation. In that case, could have been better to make a more apples-to-apples comparison even if that means using pytorch-attention for all (a separate table can be used for pytoch-attention based results). (resolved)

1. More benchmarks could have been attempted besides just the LongBench suite but not the biggest issue.


[1] Sparse Modular Activation for Efficient Sequence Modeling - Ren et al. NeurIPS 2023

[2] Learned Token Pruning for Transformers - Kim et al. KDD 2022

[2] Transkimmer: Transformer Learns to Layer-wise Skim - Guan et al. ACL 2022

**Questions:**

1. There is a lot of repeated information in the tables. It seems to me that Table 1,2,4 can be the same table with all the approaches . Just consistently show TTFT (ms), and put all the non-large models (< 70B) there (the larger models are fine on a different table).

1. The random baseline is better framed as an ablation.

1. The graph in Figure 7 seems uninterpretable if I am not missing anything. According to the text, it is supposed to show how speedup and accuracy varies with the $\eta$; but nothing the graph tells us which dot is associated with what $\eta$  or compression rates  for the other model.

1. "Unlike previous token pruning methods (Xiao et al., 2024; Li et al., 2024; Zhang et al., 2023b; Yang
et al., 2024b) where tokens are directly pruned from the whole layer, FTP dynamically selects a
certain proportion of tokens with an attention-based strategy and prunes them before the inference
of FFN." - This distinction is not sharp enough. Dynamic selection of certain proportion of tokens before FFN layers, is also direct pruning (by dynamic selection) from the whole (FFN) layers. So it's not explicitly said what the difference is. Perhaps you want to say that previously works applied token pruning to both attention + FFN, or that they apply permanent pruning (pruned in one layer => pruned in all later layers) and so on. But you need to be more explicit and sharp. .

Review Caveat: I am not as familiar with modern works on LLMs for token pruning and not entirely sure where this method stands and if the best baselines have been chosen.

---

> ### Author Response · Authors · 2024-11-22
> **Response to weaknesses and questions**
>
> Thank you for your thoughtful comments and time.
>
> **Q1. Discussion about memory impact.**
>
> **A1.** Our work focuses on the acceleration of the prefilling stage for long-context LLM inference and the memory impact of FTP is trivial. Prior KV cache eviction methods are motivated in terms of memory reduction and mainly optimize the computational and memory cost during the decoding stage, while our work reduces the computational cost during the prefilling stage.
>
> We run Qwen2-7B-Instruct with/without FTP on all datasets from LongBench, and record the peak memory usage during the prefilling stage for each data sample. The average result is in the table below.
>
> **`Peak memory usage during the prefilling stage of Qwen2-7B-Instruct`**
>
> | Dataset | Baseline (MB) | FTP (MB) | Difference (MB) |
> | :---: | :---: | :---: | :---: |
> |narrativeqa |	37262.10 |	37261.85 |	-0.25 |
> |qasper |	19887.26 |	19887.41 |	0.15 |
> |multifieldqa_en |	21728.59 |	21728.77 |	0.18 |
> |hotpotqa |	27573.94 |	27574.12 |	0.18 |
> |2wikimqa |	22077.55 |	22077.72 |	0.17 |
> |musique |	30271.91 |	30272.07 |	0.16 |
> |gov_report |	24731.25 |	24731.38 |	0.13 |
> |qmsum |	28039.78 |	28039.91 |	0.14 |
> |multi_news |	17557.00 |	17557.17 |	0.17 |
> |trec |	21406.34 |	21406.51 |	0.16 |
> |triviaqa |	26403.34 |	26403.52 |	0.18 |
> |samsum |	23598.34 |	23598.53 |	0.19 |
> |passage_count |	29404.43 |	29404.63 |	0.21 |
> |passage_retrieval_en |	27078.49 |	27078.68 |	0.18 |
> |lcc |	18039.76 |	 18039.95 |	0.19 |
> |repobench-p |	25164.82 |	 25164.99 |	0.17 |
>
> **Q2. Literature review.**
>
> **A2.** Thanks for your advice on our literature review. References [1,2,3] are significant works on token pruning outside of LLM, and we will incorporate them in the related works section in the revised version.
>
> However, the most evident differences between FTP and these works are three folds:
>
> 1. FTP focuses on inference of LLMs with billions of parameters, whereas [1,2,3] conduct experiments on models with millions to hundreds of millions of parameters;
>
> 2. FTP does not need any re-training or post-training, which is time-consuming for LLMs. However, reference [1] proposes a novel model architecture and requires training from scratch. Reference [2, 3] also involves re-training or post-training to learn some threshold or parameters to enable token pruning, which is not in the same setting as our method.
>
> 3. FTP does not prune tokens for layers and it only reduces the input tokens for the FFN module, which accounts for over 60% of the time during the prefilling stage, while [2,3] prune tokens for the whole layer or even successive layers and need re-training to preserve accuracy.
>
> **Q3. Overall trade-off feels a bit lukewarm.**
>
> **A3.** The overall trade-off of our method is considerable compared to other relevant works (e.g. LLMLingua2, PyramidInfer). As mentioned in **A1** in our *response to common questions*, PyramidInfer failed to accelerate the prefilling stage when implemented with FlashAttention. Moreover, as shown in Figure 7, FTP can achieve a larger speedup (1.4x ~ 1.5x) with bearable accuracy loss for most tasks even applied to a 7B model.
>
> For the accuracy drop by Llama3-8B-Instruct for code completion, we attribute this mainly to the limited context length (8K) of Llama3 models. To further investigate, we conduct additional experiments on Llama3.1, which supports a context length of up to 128K tokens. For a more detailed analysis and results, please refer to our response **A1** to reviewer **5w9V**.
>
> **Q4. Comparison with PyramidInfer**
>
> **A4.** Please refer to **Q1** in the *response to common questions*.
>
> **Q5. More benchmarks.**
>
> **A5.** Please refer to **Q3** in the *response to common questions*.
>
> **Q6. Regarding the paper layout.**
>
> **A6.** Thanks for your advice and we will try to merge the tables in the revised version.
>
> **Q7. The random baseline.**
>
> **A7.** Thanks for your advice, but the random baseline is already in the ablation study section (Section 4.6) if I am not misunderstanding the question.
>
> **Q8. Confusion in Figure 7**
>
> **A8.** Figure 7 illustrates the trend of accuracy as the TTFT speedup grows, and demonstrates the trade-offs between them. As mentioned in Lines 424\~425 in Section 4.4, the TTFT speedup of FTP can be modulated by tuning not only $\eta$, but $\mathcal{F}$ as well. Additionally, we also conduct specialized experiments for these two hyper-parameters in Appendix 6.1, from which most data points in Figure 7 are collected.

---

> ### Author Response · Authors · 2024-11-22
> **Response to weaknesses and questions (Cont.)**
>
> **Q9. Be more explicit and sharp.**
>
> **A9.** Thanks for your guidance on improving our manuscript and we will refine this paragraph in the revised version accordingly. As mentioned in Lines 100\~103 and depicted in Figure 4, the distinction is that FTP only prunes tokens that are input into the FFN module, yet the pruned tokens are still reserved in this layer because of the residual connection surrounding the FFN module (see the yellow tokens in Figure 4, their features are reserved before and after the FFN module). In other words, FTP does not reduce the number of tokens fed into each layer. Contrarily, previous token pruning methods applied token pruning to the whole layer (i.e. layernorm + attention + FFN), resulting in various numbers of tokens input to each layer.

---

> ### Comment · Reviewer_Pbr1 · 2024-11-23
>
> A1. Thank you for the additional information. It seems many of the prior methods that have focused on pruning and efficiency also takes memory reduction as a concern. For example even PyramidInfer seems to motivate "over 54% GPU
> memory reduction in KV cache". If your method doesn't help in that respect this should be more explicitly acknowledged as a limitation.
>
> A2. Note [1,2,3] is a non-exhaustive list. Many similar works have been attempted before - which should be ideally added to the review. They can be found from the works cited by [1,2,3] and works citing [1,2,3]. I acknowledge the differences with your work.
>
> A3. I am not sure if limited context length explains why the baseline is performing much better. Even the baseline should suffer from the effect of limited context length. But nevertheless, it's good to say the high performance drop only happened for one instance for a specific model (for whatever reason) so far.
>
> A4.  I am still not that completely convinced about the comparison. If I understand correctly, FlashAttention not returning attention weights would be simply an implementation issue not something of an in principle issue that cannot be solved in a future implementation. Recomputing attention scores again does not sound to me as a reasonable baseline. A more reasonable comparison could be running all the three methods with pytorch-implemented attention (unless you can explain why FlashAttention is theoretically impossible to utilize for PyramidInfer in an effective manner - and the bottleneck is not just accedential facts about how the implementations are currently done). Moreover, it's unclear here why only PyramidInfer faces issue for FlashAttention not returning attention weight, when your method seems to rely on attention scores too.
>
> A7. Yes, my bad.
>
> A8. My point is that dots are not annotated. For instance, I can find a dot that has speed up 1.1 and relative score 60%, and by the color I may know it's Llmlingua. But how do I know which prompt compression rate that co-ordinate represents? Trade-off graphs like that typically have annotated points.
>
> -----
>
> A4 is currently the main point that is holding me back from increasing the score.
>
> The other points should be ideally incorporated in the discussion/presentation of the paper.

---

> > ### Author Response · Authors · 2024-11-23
> >
> > Thank you for your additional comments.
> >
> > **A4.** FlashAttention not returning attention weights is exactly a principle issue. FlashAttention accelerates the attention operation mainly by reducing access to the GPU memory (because it is slow), which means that it does not write the attention weights to the GPU memory. Instead, it divides Q, K, and V into tiles, loads several tiles into the GPU SRAM, and calculates a small block of attention output at one time, and the 1) matmul between Q and K, 2) applying attention mask to $Q \cdot K^T$, 3) softmax, and 4) matmul between attn weight and V, are fused into one GPU kernel function. The attention weights and many other intermediate results are only stored temporarily in the GPU's SRAM, whose bandwidth is much larger than the GPU memory (19TB/s v.s. 1.5TB/s). In a word, not returning attention weights is one of the key designs/features of FlashAttention that makes it "flash".
> >
> > Given this background, both our method and our re-implementation of PyramidInfer with Flashattention need to re-calculate the necessary attention weights for token selection. However, our method does not need to re-calculate so many attention weights as PyramidInfer does. Please refer to **A1** in the responses to reviewer QpPZ for a detailed analysis of the computational cost of calculating the attention weights.
> >
> > Moreover, the reason why we choose FlashAttention over PyTorch-implemented Attention are two folds:
> > 1. Many popular modern LLM inference frameworks (e.g. VLLM, TensorRT-LLM) are built upon or support FlashAttention, even PyTorch2.0 announced its new function called "scaled_dot_product_attention" based on FlashAttention for faster inference, which does not return attention weights either. Our method also starts from this practical point and works better with FlashAttention.
> > 2. As shown in Table 2, previous TTFT acceleration works (e.g. PyramidInfer) based on PyTorch-implemented attention failed to outperform the baseline (standard inference) with FlashAttention. We consider LLM inference with FlashAttention a more practical setting.

---

> > > ### Comment · Reviewer_Pbr1 · 2024-11-23
> > >
> > > Thank you for the clarification. I increased my score. It would be good to add the discussion in the main paper.

---

### Official Review · Reviewer_5w9V · 2024-11-02

**Soundness:** 3
**Presentation:** 3
**Contribution:** 2
**Rating:** 6
**Confidence:** 4

**Summary:**

This paper presents a simple and effective token pruning method for FFN called FTP, designed to improve the efficiency of large language models (LLMs) during the prefilling stage of long-context inference. FTP addresses this by dynamically pruning non-critical tokens before FFN inference, using attention scores in each layer to determine token importance and pruning quantity. Unlike previous KV-cache eviction methods focusing on improving the efficiency of the decoding phase, FTP achieves speedups in TTFT with minimal performance loss.

**Strengths:**

1. This paper well-positioned its focus on the TTFT optimization of long-context LLM inference with a clear presentation of the walltime proportion of the main modules in each decoder layer during prefilling to show its motivation.
2. The FTP is simple and technically sound. Experiments on the LongBench dataset showed its effectiveness in achieving speedups in TTFT with less performance loss than baselines.
3. In the ablation study, FTP is also evaluated on larger models, which could be more practical and adopted for broad use.

**Weaknesses:**

1. There is a major performance loss on Code Completion Tasks with LLama3 model compared to baselines in Table 2. How the authors explain this phenomenon? This may impact the broader use of FTP and its effectiveness on different models.
2. In Table 2, the average performance of PyramidInfer is higher than FTP. The reported high TTFT for Pyramidinfer is not fair due to the pytorch-implemented attention. In this way, it seems hard to tell the effectiveness and efficiency of FTP compared to Pyramidinfer.

**Questions:**

1. How do the authors obtain the attention score for token pruning in FTP based on FlashAttention op?
2. Since the number of tokens with high attention scores varies among different layers, how to determine the best reserved ratio for each layer's FFN? Do the increasing number of pivotal tokens in Attention reflect that we should also reserve more tokens for FFN?

---

> ### Author Response · Authors · 2024-11-22
> **Response to weaknesses and questions**
>
> Thank you for your thoughtful comments and time.
>
> **Q1. Major performance loss on Code Completion Tasks with Llama3 model compared to baselines.**
>
> **A1.** The code completion tasks in LongBench consist of two datasets: LCC and Repobench-P. Our explanations for this phenomenon are three folds:
> 1. The insufficient context length of Llama3. Llama3 has a context length of only 8K, but the "Repobench-P" dataset has more 283 out of 500 samples that exceed the context length of Llama3, which can degrade the performance of LLMs. It is probable that the token pruning by FTP further degrades the model with insufficient context length and results in more accuracy loss.
>
> 2. On the other hand, the accuracy loss of FTP applied on Qwen2-7B, Qwen1.5-32B, and Qwen2-72B models are all acceptable, which indicates that FTP can work in various sizes of models. It is worth noting that all of these models have a context length of 32K.
>
> 3. As Llama3.1 has extended its context length to 128K, which is larger enough for the prompt length of all data samples in the "Repobench-P" and "LCC" dataset in LongBench, we evaluate FTP on Llama3.1-8B-Instruct with the same setting as Llama3-8B-Instruct in our paper,  and the results are shown below.
>
> **`Llama3.1-8B-Instruct on LongBench`**
>
> | | Single-document QA | Multi-Document QA | Summarization | Few-shot Learning | Synthetic Task | Code Completion |
> | :---: | :---: | :---: | :---: | :---: | :---: | :---: |
> | Baseline Accuracy | 43.34 | 46.02 | 28.93 | 69.49 | 52.75 | 59.85 |
> | FTP Accuracy | 42.25 | 43.05 | 26.63 | 66.08 | 54.27 | 53.71 |
> | TTFT Speedup (x) | 1.21 | 1.23 | 1.23 | 1.22 | 1.28 | 1.20 |
>
> **Q2. Comparision with Pyramidinfer.**
>
> **A2.** Please refer to Q1 in the *response to common questions*.
>
> **Q3. How to obtain the attention score based on FlashAttention?**
>
> **A3.** FlashAttention does not return attention weights. We compute the necessary attention weights (the last N rows, as depicted in Figure 4) for token selection. As we discuss in Lines 241~243 in our paper, prior work [R3] has empirically revealed that the attention pattern obtained by the queries at the end of the prompts is nearly consistent with that obtained by all queries, we only retain the attention scores M′ from the last N queries to reduce overhead.
>
> A quantitive analysis of the computational cost is in our response **A1** to reviewer **QpPZ**.
>
> **Q4. How to determine the best reserved ratio for each layer's FFN?**
>
> **A4.** The reserved ratio $\eta$ for each layer's FFN is a hyper-parameter of FTP. The optimal $\eta$ depends on the tolerance of performance degradation in the application. As mentioned in Section 4.1, for Llama3-8B-instruct, we empirically preserve the first F=10 layers and set $\eta$ = 0.90 for the following layers. When it comes to the Qwen2-7B-model, F and $\eta$ and set as 10 and 0.95 respectively. More experiments for the reserved ratio are detailed in Appendix 6.1.2.
>
> Another guidance for determining an optimal reserved ratio $\eta$ is in Section 3.2.2 and Figure 6. As depicted in Figure 6, nearly 50% of tokens in a layer are evicted when $\eta$ is set as 0.90.
>
> **Q5. Do the increasing number of pivotal tokens in Attention reflect that we should also reserve more tokens?**
>
> **A5.** Yes. The number of reserved tokens is determined by the hyper-parameter $\eta$ for this layer. If there are more pivotal tokens (i.e. more tokens with high attention scores), our method would adaptively reserve more pivotal tokens to achieve the percentage ($\eta$) of the summed attention weights.

---

> > ### Comment · Reviewer_5w9V · 2024-11-25
> >
> > Thanks for the authors' clarification.
> >
> > As the author mentioned in the common response for Q1:  *PyramidInfer can be implemented with FlashAttention by re-calculating the necessary part of the attention weights for token selection. However, the re-calculation is computationally expensive.*
> >
> > The re-calculation procedure is also conducted in FTP, so I wonder why the Pyramidinfer costs much more than FTP.
> >
> > Can the Pyramidinfer also utilize the last N queries to retrain the attention score for faster speed? I would like to see the accuracy and efficiency comparison in this circumstance.

---

> > > ### Author Response · Authors · 2024-11-29
> > >
> > > Thanks for your comments and patience.
> > >
> > > The reason why PyramidInfer costs much more than FTP is that they retain 20% of attention weights for token pruning while FTP just utilize the last N=50 queries to calculate the attention weights.
> > >
> > > We have re-implemented PyramidInfer to utilize the last N queries to calculate the attention weights and here are the results on Llama3-8B-Instruct.
> > >
> > > **`Accuracy Score and TTFT Speedup on LongBench with Llama3-8B-Instruct`**
> > >
> > > | Tasks | Method | Score | TTFT Speedup |
> > > | :---: |  :---: |  :---: |  :---: |
> > > | Single-Document QA | Baseline | **37.20** | 1.00 |
> > > | Single-Document QA | PyramidInfer | 36.41 | **1.22** |
> > > | Single-Document QA | Ours | 36.06 | 1.20 |
> > > | | | | |
> > > | Multi-Document QA | Baseline | **36.85** | 1.00 |
> > > | Multi-Document QA | PyramidInfer | 35.22 | 1.17 |
> > > | Multi-Document QA | Ours | 34.85 | **1.21** |
> > > | | | | |
> > > | Summarization | Baseline | **26.80** | 1.00 |
> > > | Summarization | PyramidInfer | 25.14 | **1.26** |
> > > | Summarization | Ours | 24.41 | 1.19 |
> > > | | | | |
> > > | Few-shot Learning | Baseline | **69.33** | 1.00 |
> > > | Few-shot Learning | PyramidInfer | 67.85 | 1.16 |
> > > | Few-shot Learning | Ours | 67.55 | **1.21** |
> > > | | | | |
> > > | Synthetic | Baseline | 37.00 | 1.00 |
> > > | Synthetic | PyramidInfer | **37.50** | 1.15 |
> > > | Synthetic | Ours | 36.00 | **1.25** |
> > > | | | | |
> > > | Code Completion | Baseline | 55.17 | 1.00 |
> > > | Code Completion | PyramidInfer | **55.43** | **1.29** |
> > > | Code Completion | Ours | 35.91 | 1.19 |
> > >
> > > As shown in the results, our method still have comparable accuracy scores with PyramidInfer and achieves higher TTFT speedup for the multi-document QA, few-shot learning and synthetic tasks. For the accuracy drop in the code completion tasks, we have discussed with you in **A1**.
> > >
> > > Moreover, the selection of N is a crucial aspect of our design for practical inference, which we extensively explored in our experimental analysis presented in Section 6.1.3, whereas PyramidInfer does not take into account the scenario of Flash Attention nor investigates the correlation between the "recent length" and the inference speed.

---

> > > > ### Comment · Reviewer_5w9V · 2024-12-02
> > > >
> > > > Thanks for the authors' clarification. I have increased my score to 6.

---

### Official Review · Reviewer_CwS7 · 2024-11-02

**Soundness:** 3
**Presentation:** 2
**Contribution:** 3
**Rating:** 5
**Confidence:** 4

**Summary:**

This paper proposes a token pruning method called FTP to accelerate the prefill stage inference of long-context LLMs. The authors first demonstrate the importance of accelerating the prefill stage, noting that it accounts for over 60% of the inference time (it should be noted that this is the case when the number of tokens to be decoded is around 10~20). They identify the FFN module as the most time-consuming component. Based on this, FTP selects non-critical tokens from the input sequence at each layer using attention scores and omits these tokens from the FFN computation. In the experimental section, the authors use results from Longbench to demonstrate that FTP can accelerate the prefill stage while maintaining a certain level of performance.

**Strengths:**

- The idea of accelerating the prefill stage by pruning tokens in the FFN layer is innovative.
- The paper provides a clear explanation of the motivation and methodology.

**Weaknesses:**

- The experimental section is relatively weak: 1) The benchmarks tested are limited, with only Longbench included. It is recommended to add another benchmark, such as BigBench, ZeroScrolls, or L-Eval; 2) The baselines tested are limited, with only PyramidInfer and LLMLingua2 included. The authors should add more baselines or explain why only these two baselines were selected.
- Some parts are not clearly written. For example, in lines 232-236, it is unclear how conclusions 2 and 3 are derived from Figure 5.

**Questions:**

- Typo: line 267: "and the dataset" should be "dataset".
- If the authors can address the issues raised in the weaknesses section, I would consider increasing the score.

---

> ### Author Response · Authors · 2024-11-22
> **Response to weaknesses and questions**
>
> Thank you for your thoughtful comments and time.
>
> **Q1. Add another benchmark**
>
> **A1.** Please refer to **Q3** in the response to common questions.
>
>
>
> **Q2. Why only these two baseline are selected**
>
> **A2.** Please refer to **Q2** in the response to common questions.
>
>
> **Q3. How conclusions 2 and 3 are derived from Figure 5.**
>
> **A3.** Let us recall conclusion 2 and 3 first.
> > Conclusion2: tokens with high attention scores in one layer may not necessarily be prioritized in the same manner across other layers;
>
> > Conclusion3: the number of tokens with high attention scores varies among different layers.
>
> Conclusion2 can be inferred by analyzing the color of a single column in a sample in Figure 5. For example, the token at position 0 exhibits high attention scores in layers 1\~6 and 16\~28, yet its scores decrease in layers 10\~15 and layer 31. A similar pattern is distinctly observable on the tokens at positions larger than 100. Moreover, tokens at position 60\~80 also support our conclusion, even though the variation in attention scores is not so obvious as that of the tokens at the initial and final positions. Based on this conclusion, FTP is designed to select important tokens in each layer.
>
> Conclusion3 can be inferred by analyzing the color of a single row in a sample in Figure 5. For instance, in layers 5\~18, there are more tokens with high attention scores, and in layers 2\~3 or layers 20\~25, the attention scores highly concentrate on the initial tokens, resulting in fewer high-score tokens. This conclusion motivates us that the number of tokens to be retained should be determined by a ratio $\eta$ of attention scores, instead of an absolute number.
>
> **Q4. Typo in Line 267**
>
> **A4.** Thanks for your advice and we will correct it in the revised version.

---

> > ### Comment · Reviewer_CwS7 · 2024-11-23
> > **Reply to the authors**
> >
> > Dear authors,
> >
> > Thanks for your reply. I'm glad to see that you have provided explanations for choosing baselines, and for the conclusions derived from Figure 5. I'm looking forward to the results on L-Eval.

---

> > > ### Author Response · Authors · 2024-12-02
> > > **Results on L-Eval**
> > >
> > > Dear Reviewer CwS7,
> > >
> > > We have conducted experiments on the L-Eval benchmark and the results are in Section 6.1.1 of the revised version. Here is a brief introduction to our experiments and the results.
> > >
> > > L-Eval has two groups of tasks: closed-ended tasks and open-ended tasks. The former group focus on evaluating an LLM’s ability of reasoning and understanding a long context, while the latter one is more about summarization tasks for a long document. We run FTP and PyramidInfer on the Llama3-8B-Instruct and Qwen2-7B-Instruct models for both tasks, and the results are in the tables below.
> > >
> > > **`L-Eval results on closed-ended tasks. "TRL" stands for the "topic retrieval longchat" dataset, and "SpeedUp" represents the TTFT speedup w.r.t. the baseline`**
> > >
> > > | Model |  Coursera |  GSM |  QuALITY |  CodeU | SFiction |  TRL |  TPO | Avg.  |  SpeedUp |
> > > |:---:|:---:|:---:|:---:|:---:|:---:|:---:|:---:|:---:|:---:|
> > > | Llama3 Baseline |  52.76 |  66.00 |  **60.40** |  **4.44** | **71.09** |  **62.00** |  75.46 | **56.02**  |  1.00 |
> > > | Llama3 + FTP |  **53.20** |  38.00 |  55.45 |  4.44 | 69.53 |  52.00 |  74.35 | 49.57  |  1.21 |
> > > | Llama3 + PyramidInfer |  52.18 |  **68.00** |  60.40 |  4.44 | 71.09 |  49.33 |  **75.84** |  54.47 |  **1.24** |
> > > |||||||||
> > > | Qwen2 Baseline | 68.60 |  24.00 |  **65.35** |  7.78 | **73.44** |  **47.33** |  **81.04** |  52.51 |  1.00 |
> > > | Qwen2 + FTP |  68.60 |  **30.00** |  65.35 |  **10.00** | 72.66 |  43.33 |  80.30 |  **52.89** |  **2.50** |
> > > | Qwen2 + PyramidInfer |  **69.77** |  30.00 |  65.35 |  6.67 | 71.88 |  42.67 |  81.04 |  52.48 |  2.01 |
> > >
> > > ---
> > >
> > > **`L-Eval results on open-ended tasks. The output of models are compared with that of GPT3.5-turbo-16k-0613 by the judge model (i.e., GPT4).`**
> > > | Model | Wins | Ties | Win-rate (%) | TTFT SpeedUp |
> > > |:---:|:---:|:---:|:---:|:---:|
> > > | Llama3 Baseline | 50 | 62 | **42.19** | 1.00 |
> > > | Llama3 + FTP | 44 | 67 | 40.36 | **1.19** |
> > > | Llama3 + PyramidInfer | 49 | 51 | 38.80 | 1.08 |
> > > ||||
> > > | Qwen2 Baseline | 42 | 75 | 42.97 | 1.00 |
> > > | Qwen2 + FTP | 47 | 69 | **43.58** | **1.21** |
> > > | Qwen2 + PyramidInfer | 35 | 69 | 37.37 | 1.09|
> > >
> > > For closed-ended tasks, FTP with the Llama3 model sustains considerable performance across half of the datasets, and achieves a TTFT speedup comparable to PyramidInfer. When applied on Qwen2-7B-Instruct model, FTP not only enhances the performance on the GSM (Cobbe et al., 2021) and CodeU datasets, but also surpasses the baseline scores. Furthermore, the longer context length for Qwen2 illustrates the acceleration of FTP during the prefilling stage, showcasing its efficiency.
> > >
> > > For open-ended tasks, we employ GPT4 as a judge to compare the outputs from the model with those from GPT-3.5-turbo-16k-0613. Each pair of predictions is evaluated by GPT4 twice, with their positions swapped after the first judgment. The number of wins and ties for the model are recorded in the table. As illustrated in the results, FTP achieves higher TTFT speedup for both models, and notably, the performance on Qwen2-7B-Instruct even surpasses that of the baseline.

---

### Official Review · Reviewer_QpPZ · 2024-11-03

**Soundness:** 2
**Presentation:** 3
**Contribution:** 3
**Rating:** 5
**Confidence:** 5

**Summary:**

1. The paper introduces a method called FFN Token Pruning (FTP) to optimize the inference phase of large language models (LLMs) or long-context processing by selectively pruning tokens during FFN computations.
2. FTP dynamically prunes tokens with low attention weights.
3. FTP achieves faster inference speeds, showing up to a 1.39x improvement on models like Qwen2-7B-Instruct.

**Strengths:**

1. The method is straightforward and easy to apply.
2. The experiments demonstrate strong performance results.
3. The experimental setup is well-explained and easy to understand.
4. The analysis of experimental results is detailed and thorough.

**Weaknesses:**

1. FTP involves a considerable computational cost in calculating the Sum Attention Score, but this issue is not discussed by the authors.
2. FTP applies KVCache Eviction (Compression) methods to FFN token pruning without justification or evidence that the Attention Score is also effective for FFN token pruning. Additionally, FTP lacks a specific design tailored to FFN computations.
3. In Figure 5, FTP appears to retain only the beginning and end tokens, discarding most intermediate information. However, previous studies have shown that losing intermediate information can lead to sampling bias and performance loss [1].
4.  Similar research that also performs FFN token pruning, such as MoD [2], is not discussed in the paper. The choice of comparison methods is limited; for example, LongBench compares only with LLMLingua2 and lacks comparison with methods like KVCache Eviction, such as H2O [3] and NACL [1]. When compared with PyramidInfer, the setup lacks fair consistency, as the FlashAttention setting was not standardized.


---

References:
[1] Y. Chen et al., “NACL: A General and Effective KV Cache Eviction Framework for LLMs at Inference Time.”

[2] D. Raposo, S. Ritter, B. Richards, T. Lillicrap, P. C. Humphreys, and A. Santoro, “Mixture-of-Depths: Dynamically allocating compute in transformer-based language models.”

[3] Z. Zhang et al., “H₂O: Heavy-Hitter Oracle for Efficient Generative Inference of Large Language Models.”

**Questions:**

1. Could you explain why the Attention Score is effective for FFN token pruning?
2. Could you discuss the differences between this work and other dynamic layer-skipping methods?
3. Please analyze the eviction effects of FTP and assess whether it may result in a loss of intermediate information.

---

> ### Author Response · Authors · 2024-11-22
> **Response to weaknesses and questions**
>
> Thank you for your thoughtful comments and time.
>
> **Q1. Regarding the computational cost in calculating the Sum Attention Score.**
>
> **A1.** The computational cost in calculating the sum attention score is trivial since we only calculate the attention scores from the last N queries to all input keys (referenced in Line 180 in our paper) instead of calculating the full attention scores. The value of N is much smaller than the input length L. We set N = 50 in our experiments, which means that N is 2\~3 orders of magnitude lower than L in most long-context scenarios. Furthermore, we conduct an experiment on LongBench with Llama3-8B-Instruct and Qwen2-7B-Instruct and summarize the TTFT of FTP and the random variant (referenced in Section 4.6.1), which drops the same number of tokens in each layer as FT with negligible computational cost. As shown in the table below, FTP on Llama3-8B-Instruct only introduces a little computational cost of about 7\~10ms, which accounts for only 1%\~3% of the TTFT, whereas FTP on Qwen2-7B-Instruct introduces a cost of 8\~15ms, which takes up 0.8%\~1.9% of the TTFT.
>
> **`TTFT (ms) of Llama3-8B-Instruct`**
> | | Single-document QA | Multi-Document QA | Summarization | Few-shot Learning | Synthetic Task | Code Completion |
> | :---: | :---: | :---: | :---: | :---: | :---: | :---: |
> | FTP | 480.24 | 551.23 | 438.08 | 489.26 | 584.18 | 375.22 |
> | Random Drop | 469.99 | 541.11 | 428.95 | 481.82 | 573.84 | 367.09 |
> | Extra cost of FTP | 10.25 | 10.12 | 9.13 | 7.44 | 10.35 | 8.13 |
> | Proportion | 2.18% | 1.87% | 2.13% | 1.54% | 1.80% | 2.22% |
>
> **`TTFT (ms) of Qwen2-7B-Instruct`**
> | | Single-document QA | Multi-Document QA | Summarization | Few-shot Learning | Synthetic Task | Code Completion |
> | :---: | :---: | :---: | :---: | :---: | :---: | :---: |
> | FTP | 923.18 | 859.11 | 647.59 | 641.58 | 955.94 | 498.15 |
> | Random Drop | 907.75 | 845.63 | 636.32 | 630.81 | 947.67 | 489.04 |
> | Extra cost of FTP | 15.43 | 13.48 | 11.27 | 10.77 | 8.27 | 9.12 |
> | Proportion | 1.70% | 1.59% | 1.77% | 1.71% | 0.87% | 1.86% |
>
> **Q2. Justification or evidence that the Attention Score is also effective for FFN token pruning. Lacks design tailored to FFN.**
>
> **A2.** First of all, FTP does not evict tokens as the KV Cache eviction methods do. FTP allows pruned tokens to retain their features and pass them to the next layer. The insight of using attention scores for FFN token pruning is that tokens with low attention scores in the current layer are very likely to have low attention scores in the next layer (see the columns in the samples in Figure 5 and Appendix 6.2). Therefore, these tokens can bypass updates by the FFN module and simply retain their features from the attention module in the current layer. As these features are fed into the attention module in the next layer, the error introduced by omitting the FFN update for the pruned tokens is minimized since their attention scores are still low.
>
> Moreover, by design, we manually preserve the first P tokens and the last N tokens in each layer, ensuring that they are never pruned,  since their attention scores in the current layer are not always consistent with that of the next layer (see the columns in the samples in Figure 5).
>
> Last but not least, our experiment in section 4.6.1 is also evidence of the effectiveness of our attention-based strategy.
>
> One of our specific designs for FFN is that we utilize the residual connection to retain more information of the pruned tokens instead of discarding them as in the KV Cache eviction methods. Another design is that we logically set the FFN outputs of the pruned tokens to zero vectors. As depicted in the "Feed-forward Network" part in Figure 4,  if the feature of a token is deactivated after the "Activation" module (e.g. the SiLU function), its FFN output closely resembles a zero vector. Hence, the error introduced by skipping the FFN module can be reduced by setting the FFN outputs of the pruned tokens to zero vectors.
>
> **Q3. FTP discards most intermediate tokens. Please analyze the eviction effects of FTP and assessment of the loss of intermediate information.**
>
> **A3.** On one hand, FTP does not discard tokens. As mentioned in Lines 102~103, FTP prunes tokens that are fed into FFN. However, the pruned tokens are just not updated by the residual produced by the FFN, and their feature remains unchanged due to the residual connection surrounding the FFN module. On the other hand, as illustrated in Figure 6, FTP still reserves nearly 60% of tokens when $\eta$ is set to 0.95, which indicates that not only the beginning and end tokens are retained. We further visualize the distribution of the relative position of the retained tokens in the figure in the revised version.
>
> We are also running "needle in a haystack" [R5] experiments for FTP, which provides a quantitative assessment of the loss of intermediate information.
>
> ---
> References
> > [R5] G Kamradt. Needle in a haystack–pressure testing llms, 2023.

---

> ### Author Response · Authors · 2024-11-22
> **Response to weaknesses and questions (Cont.)**
>
> **Q4. Similar research that also performs FFN token pruning, such as MoD[2], is not discussed in the paper.**
>
> **A4.** Thanks for your suggestions on our literature review, and we will include MoD in our related works. MoD applies a router to determine for a token whether to skip **both** the self-attention and MLP blocks in a layer, and it needs training for the model. On the other hand, FTP only prunes tokens for the FFN module and does not need any post-training for the LLM, which is time-consuming. In the experiment section, MoD is applied to models with 60M to 3B parameters, whereas FTP focuses on the inference of LLMs and incorporates models with 7B~72B parameters.
>
> **Q5. The choice of comparison methods.**
>
> **A5.** Please refer to **Q2** in the *response to common questions*.
>
> **Q6. Inconsistency between the attention implemetation of PyramidInfer and FTP.**
>
> **A6.** Please refer to **Q1** in the *response to common questions*.
>
> **Q7. Discuss the difference between this work and other dynamic layer-skipping methods**
>
> **A7.** The most significant difference between FTP and other layer-skipping methods is that FTP only reduces the number of tokens fed into the FFN, and **the number of tokens fed into each layer remains the same** across all layers, since the pruned tokens remain identical after the residual connection. Some layer-skipping methods [2, R6] require post-training which is time-consuming for LLMs.
>
> ---
> References
>
> > R6. Elhoushi, M., Shrivastava, A., Liskovich, D., Hosmer, B., Wasti, B., Lai, L., ... & Wu, C. J. (2024). Layer skip: Enabling early exit inference and self-speculative decoding. arXiv preprint arXiv:2404.16710.

---

### Author Response · Authors · 2024-11-22
**Response to common questions**

**Q1. Comparison with Pyramidinfer**

+ PyramidInfer achieves considerable acceleration only with PyTorch-implemented attention, since it requires a large part of attention weights for token selection.
+ FlashAttention cannot return attention weights.
+ The PyTorch-implemented attention is much slower than FlashAttention, and is not a practical choice for modern LLM inference. Table 2 in our paper shows that even the optimized inference (PyramidInfer) with PyTorch-implemented attention is obviously slower than the FlashAttention baseline.
+ PyramidInfer can be implemented with FlashAttention by re-calculating the necessary part of the attention weights for token selection. However, the re-calculation is computationally expensive.
+ We re-implement PyramidInfer with **FlashAttention** and conduct experiments with Llama3-8B-Instruct on LongBench. As shown in the table below, PyramidInfer can hardly reduce TTFT when implemented with FlashAttention.
+ We will supplement the experimental results below to Table 2 to avoid confusion.

**`Accuracy`**
| | Single-document QA | Multi-Document QA | Summarization | Few-shot Learning | Synthetic Task | Code Completion |
| :---: | :---: | :---: | :---: | :---: | :---: | :---: |
| Baseline | 37.20 | 36.85 | 26.80 | 69.33 | 37.00 | 55.17 |
| PyramidInfer | 32.05 | 33.43 | 24.21 | 66.37 | 35.50 | 55.24 |
| FTP | 36.06 | 34.85 | 24.41 | 67.55 | 36.00 | 35.91 |

**`TTFT SpeedUp`**
| | Single-document QA | Multi-Document QA | Summarization | Few-shot Learning | Synthetic Task | Code Completion |
| :---: | :---: | :---: | :---: | :---: | :---: | :---: |
| Baseline | 1.00 | 1.00 | 1.00 | 1.00 | 1.00 | 1.00 | 1.00 |
| PyramidInfer | 1.02 | 0.97 | 1.06 | 0.97 | 0.94 | 1.10 |
| FTP | 1.20 | 1.21 | 1.19 | 1.21 | 1.25 | 1.19 |

**Q2. The reason for selecting these two baselines**

In this work, we focus on accelerating the prefilling stage for long-context LLM inference without post-training. Since there are few works on this setting, we only choose these two baselines: 1) PyramidInfer, a pioneer work in compressing KV Cache in both the prefilling stage and the decoding stage; 2) LLMLingua2, a SOTA prompt compression method which can also accelerate the prefilling stage by reducing the length of the input prompt.

Other KV Cache Eviction methods, such as H2O [R1], NACL [R2], and SnapKV [R3], mainly focus on the reduction of memory footprint and the inference time in the decoding stage, which brings zero / negative acceleration for the prefilling stage and thus has no impact on TTFT or even increases TTFT. These works are summarized in the "Related Works" section in our paper and are not suitable for comparison in the experiment section.

**Q3. Limited benchmarks**

LongBench is a multi-task benchmark for a comprehensive assessment of the long context understanding capabilities of LLMs. Our method has been evaluated on all 16 datasets in LongBench across 6 key long-text application scenarios: single-document QA, multi-document QA, summarization, few-shot learning, synthetic tasks and code completion.

We are running more experiments on the L-Eval [R4] benchmark, which is another long-context language model benchmark. We will supplement the results once the experiments are finished.

---
References
> R1. Zhang, Z., Sheng, Y., Zhou, T., Chen, T., Zheng, L., Cai, R., ... & Chen, B. (2023). H2o: Heavy-hitter oracle for efficient generative inference of large language models. Advances in Neural Information Processing Systems, 36, 34661-34710.

> R2. Chen, Y., Wang, G., Shang, J., Cui, S., Zhang, Z., Liu, T., ... & Wu, H. (2024). Nacl: A general and effective kv cache eviction framework for llms at inference time. arXiv preprint arXiv:2408.03675.

> R3. Li, Y., Huang, Y., Yang, B., Venkitesh, B., Locatelli, A., Ye, H., ... & Chen, D. (2024). Snapkv: Llm knows what you are looking for before generation. arXiv preprint arXiv:2404.14469.

> R4. An, C., Gong, S., Zhong, M., Zhao, X., Li, M., Zhang, J., ... & Qiu, X. (2023). L-eval: Instituting standardized evaluation for long context language models. arXiv preprint arXiv:2307.11088.

---

### Author Response · Authors · 2024-11-28
**Modifications of the revised manuscript**

Dear Reviewers,

We sincerely appreciate your constructive comments and suggestions. We have made several modifications to our manuscript. Here are a summary of them:

Major modifications
- We merged the results for PyramidInfer to Table 1, and supplemented the explanations for the attention implementations in Section 4.1 (Implementation Details) and Section 4.3 (i.e., Comparison with State-of-the-art Method).
- New results for "L-Eval" and "Needle-in-a-Haystack" have been included in Section 6.1.1 and Section 6.1.2 respectively.
- We merged the results in A1 of reviewer QpPZ into Table 3 to analyse the computational cost introduced by FTP.

Minor modifications
- We incorporated more related works on layer skipping and token pruning out of LLMs.
- We fixed the typo in Line 267 as pointed out.

Thank you for your valuable feedback and for the time you have invested in reviewing our work.

Best Regards,

Authors

---

### Meta-Review · Area_Chair_n7fW · 2024-12-21

**Metareview:**

The paper proposes a token pruning method called FFN Token Pruning that accelerates the prefilling stage of long-context LLM inference. FTP dynamically determines token importance based on attention scores and prunes non-critical tokens before they are processed by the FFN. While generally effective, FTP shows significant performance degradation on code completion tasks with the Llama3 model in the initial experiments, raising concerns about the method's robustness and generalizability across different models and tasks. Other relevant methods (especially on dynamic layer skipping) could have been included for a more comprehensive comparison. Given the accuracy trade-offs in some cases, the practical benefit of this speedup might not be substantial enough to justify widespread adoption, especially considering the complexity it adds to the inference pipeline. The paper could be significantly strengthened by addressing these weaknesses in future work.

**Additional Comments On Reviewer Discussion:**

The initial evaluation was primarily on LongBench. While a comprehensive benchmark, adding other benchmarks provides a more robust evaluation.

---

### Decision · Program_Chairs · 2025-01-22

Reject